# LLM Strategic Reasoning: Agentic Study through Behavioral Game Theory

**Jingru Jia**[*]**, Zehua Yuan**[*]**, Junhao Pan, Paul E. McNamara, and Deming Chen**

University of Illinois at Urbana-Champaign
{jingruj3, zehuay2, jpan22, mcnamar1, dchen}@illinois.edu

## Abstract

What does it truly mean for a language model to "reason" strategically, and can scaling up alone guarantee intelligent, context-aware decisions? Strategic decision-making requires adaptive reasoning, where agents anticipate and respond to others' actions under uncertainty. Yet, most evaluations of large language models (LLMs) for strategic decision-making often rely heavily on Nash Equilibrium (NE) benchmarks, overlook reasoning depth, and fail to reveal the mechanisms behind model behavior. To address this gap, we introduce a behavioral game-theoretic evaluation framework that disentangles intrinsic reasoning from contextual influence. Using this framework, we evaluate 22 state-of-the-art LLMs across diverse strategic scenarios. We find models like GPT-o3-mini, GPT-o1, and DeepSeek-R1 lead in reasoning depth. Through thinking chain analysis, we identify distinct reasoning styles—such as maximin or belief-based strategies—and show that longer reasoning chains do not consistently yield better decisions. Furthermore, embedding demographic personas reveals context-sensitive shifts: some models (e.g., GPT-4o, Claude-3-Opus) improve when assigned female identities, while others (e.g., Gemini 2.0) show diminished reasoning under minority sexuality personas. These findings underscore that technical sophistication alone is insufficient; alignment with ethical standards, human expectations, and situational nuance is essential for the responsible deployment of LLMs in interactive settings.

## 1 Introduction

The rapid development of large language models (LLMs) and generative AI has significantly broadened their applications, transitioning from basic text generation (41) and completion tasks to serving as sophisticated agents (45; 60; 21; 17). While existing benchmarks primarily assess LLMs on isolated language or math tasks (33; 23; 58), real-world deployment often demands more complex forms of decision-making, particularly strategic reasoning, where agents must interact with other entities whose actions directly influence the outcomes (64; 31; 22). Specifically, one-shot strategic reasoning refers to an agent's ability to select a single, irreversible action where the outcome depends on both its own choice and that of other agents in this interaction. Consider an AI agent operating in a business environment where strategic interaction is critical. In procurement, for instance, it must anticipate supplier counteroffers when negotiating orders; in advertisement bidding, it predicts competitors' bids to set optimal prices. In each case, the agent faces a one-shot decision that directly impacts its payoff and cannot be revised post-execution (43; 12).

**Related Literature and Research Gap.** Research on LLM decision-making typically begins by examining their behavior as independent decision-makers (4; 11; 34; 16). From an economic perspective, LLMs exhibit human-like patterns in preferences under uncertainty and risk (40). In social science, they have demonstrated alignment with human responses in moral judgment and

---

[1]* Equal contribution

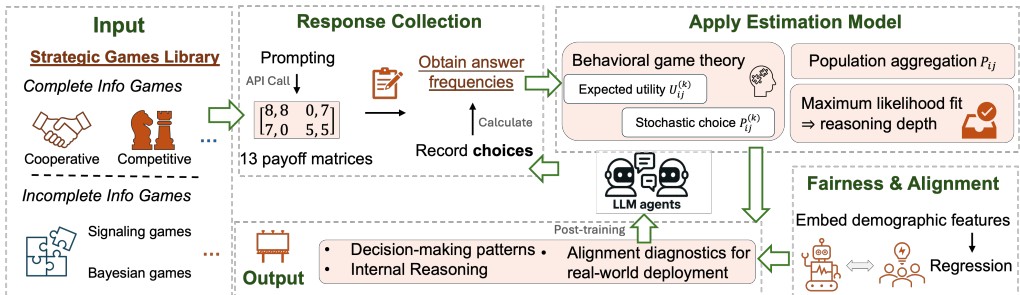

Figure 1: Framework overview

fairness-related tasks (15; 39; 52). Other studies further reveal alignments between LLMs and human judgments across diverse decision contexts (34; 46; 40). However, when demographic profiles are embedded into prompts, systematic inconsistencies and fairness concerns emerge (29; 40).

Strategic decision-making extends individual reasoning into interactive contexts, where outcomes depend not only on one's own choices but also on the anticipated actions of others. Existing evaluations of LLM strategic reasoning are predominantly grounded in game theory. Several studies explore LLMs' abilities in achieving rationality within game-theoretic frameworks, such as mixed-strategy Nash Equilibrium (NE) games (53) and classic strategic games like the prisoner's dilemma (5; 28). Some works investigate the adaptability of LLMs to various prompts that influence cooperative and competitive behavior (50), and others evaluate their capacity to replicate human-like reasoning patterns in multi-agent settings (1; 61). While these studies provide a good starting point, many are limited to binary assessments of whether or not LLMs meet NE (35; 18) without quantifying their strategic reasoning capabilities or exploring the underlying mechanisms. While NE is a cornerstone concept in evaluating rational behavior, evaluating solely whether LLMs can achieve NE is an incomplete assessment of their reasoning capabilities (36; 32). The primary limitation of using NE as a measurement lies in its strict assumptions and focus on outcomes rather than the decision-making processes (3). It makes it difficult to interpret why LLMs deviate from optimal strategies. Also, NE assumes perfect rationality, which fails to account for the variability and bounded reasoning capability inherent in real-world problems. For LLMs—probabilistic models trained on human-generated data—this assumption is particularly problematic (63; 42). Their stochastic nature makes NE impractical as a comprehensive evaluation metric, and its assumption of perfect rationality falls into the circular reasoning fallacy when assessing LLMs' rationality. Recent work begins to address these issues by incorporating stepwise evaluation (19) or applying models like Cognitive Hierarchy (CH) (65), but these approaches still rely on deterministic assumptions about context and agent type.

**Key Contributions.** To bridge this gap, we propose a framework, as shown in Figure 1, that moves beyond assessing LLMs with traditional game theory. We develop a method to quantify and characterize LLMs' strategic decision-making, capturing bounded rationality, response to incentives, and belief formation. Our approach combines a diverse library of matrix games with a structured modeling framework based on Truncated Quantal Response Equilibrium (TQRE) stemming from behavioral game theory (51; 59). The key contributions are as follows:

1. We introduce an evaluation framework for assessing strategic reasoning in LLMs under various conditions. Applying to 22 state-of-the-art (SOTA) models on 13 abstracted real-world games, we find that massive models such as GPT-o3-mini and DeepSeek-R1 achieve top reasoning depth across most tasks, while smaller models occasionally match or even outperform them in specific game types.

2. Based on our evaluation results, we investigate why different models exhibit varying reasoning depth across games by analyzing the reasoning chains of three top-performing models in baseline competitive, cooperative, and mixed-motive games. Analysis reveals that each model's performance is closely tied to its dominant reasoning style, but does not benefit well from long reasoning chains.

3. Finally, we examine the social effects and alignment of LLMs relative to human behavior when demographic features are assigned. We find that embedding demographic attributes prior to reasoning can reveal biases, even in advanced models such as DeepSeek-R1 and GPT-4o, particularly when processing minority-related features. This indicates that superior reasoning capabilities do not inherently lead to desirable or ethical outcomes, underscoring the need for careful calibration and a balanced approach in future LLM development.

## 2 Theoretical Foundations

### 2.1 Literature on Behavioral Game Theory

Classical game theory begins with the formalism of NE, in which subjects are assumed to be fully rational: they possess unlimited reasoning capacity, have perfect knowledge of the game, and make deterministic best responses (48). While mathematically sound, empirical evidence demonstrates frequent deviations from NE. For example, subjects often exhibit bounded rationality and stochastic behavior, such as overbidding in auctions or early cooperation in public goods games (25; 20). This gap between theory and observed behavior prompted the emergence of behavioral game theory, which progressively relaxes NE's strong assumptions (7; 9; 8; 54).

The first core insight is that subjects do not always select the utility-maximizing option deterministically. Instead, they respond to incentives in a probabilistic pattern, leading to stochastic choice models, as proposed in the Quantal Response Equilibrium (QRE) model (44; 30). Another observation is that subjects differ in their depth of strategic reasoning levels. Rather than assuming that all players engage in unbounded thinking, models such as CH posit that individuals operate at varying levels of sophistication (54; 9). To capture both bounded reasoning and probabilistic behavior, we adopt the TQRE model, which combines the key behavioral refinements of QRE and CH. Instead of assuming convergence to a Nash fixed point, TQRE models subjects as reasoning at limited depths and responding stochastically, thus capturing more realistic strategic behavior, structured as below.

### 2.2 Modeling Preliminaries

**Bounded-rational Belief Formation.** Each subject draws a reasoning level $k \sim \text{Poisson}(\tau)$, $\tau > 0$, where $\tau$ is the average depth of strategic thought. A level-$k$ subject does not know each opponent's exact level $h < k$, so it mixes their level-$h$ strategy distributions $\sigma_j^{(h)} : A_j \to [0,1]$, $\sum_{a_j} \sigma_j^{(h)}(a_j) = 1$, using weights $W_h^{(k)}$, to form the weighted average of lower-level strategies. The marginal belief is

$$\bar{\sigma}_j^{(k)}(a_j) = \sum_{h=0}^{k-1} W_h^{(k)} \, \sigma_j^{(h)}(a_j).$$

Given that opponents act independently, the joint belief over all others' actions $a_{-i} \in A_{-i}$ is

$$\Pr(a_{-i}) = \prod_{j \neq i} \bar{\sigma}_j^{(k)}(a_j)$$

, where the notation $a_{-i}$ stands for the actions of all players except $i$.

**Expected Utility (EU) and Stochastic Choice.** Given $\Pr(a_{-i})$, a level-$k$ subject's expected payoff for each action $a_{ij} \in A_i$ is

$$U_{ij}^{(k)} = \sum_{a_{-i} \in A_{-i}} \Pr(a_{-i}) \, u_i(a_{ij}, a_{-i})$$

, where $u_i$ is the payoff function. To allow for decision noise, set $\lambda_k = \gamma k$ with $\gamma > 0$, and convert utilities into choice probabilities:

$$p_{ij}^{(k)} = \frac{\exp(\lambda_k \, U_{ij}^{(k)})}{\sum_{a \in A_i} \exp(\lambda_k \, U_{ia}^{(k)})}.$$

**Population-Level Aggregation.** Finally, the overall probability that a randomly selected subject plays $a_{ij}$ is the mixture over reasoning levels:

$$p_{ij} = \sum_{k=0}^{\infty} f_k \, p_{ij}^{(k)}, \quad f_k = \frac{\tau^k \, e^{-\tau}}{k!},$$

By estimating $\tau$, we measure the reasoning depth LLMs entertain when making an interactive decision. We thus recover a cognitively grounded metric of bounded strategic reasoning that moves beyond NE-based assumptions to characterize how LLMs reason in multi-agent environments. For more details of the model specification, see the Appendix A.

Our modeling approach offers two key advantages over traditional NE-based evaluations. First, by incorporating bounded rationality and heterogeneous reasoning levels, it avoids the circularity of

assuming optimal behavior to test for rationality. Instead, it enables empirical estimation of strategic depth without presupposing NE play. Second, it captures the anticipatory nature of strategic reasoning: each agent's EU is grounded in beliefs about opponents' actual behavior, where $\tau$ quantifies the model's depth of recursive belief modeling, i.e., layers of back-and-forth reasoning about others' reasoning it engages in. This provides a cognitively meaningful measure of interactive sophistication.

**Interpreting $\gamma$ (decision precision).** In the logit choice rule, $\gamma$ scales how deterministically utilities are translated into actions conditional on beliefs. Larger $\gamma$ yields more deterministic (near–best-response) behavior, while lower $\gamma$ reflects noisier or more exploratory choice behavior. Importantly, $\gamma$ does not encode preferences over outcomes—it captures the consistency of choice given utilities and beliefs. Because $\gamma$ can be context- and scale-dependent, we report it primarily to avoid conflating shallow reasoning with stochastic execution, while using $\tau$ as our core measure of strategic sophistication. Full $\gamma$ estimates are reported in Appendix B.

## 3 Experiment Setup and Estimation Framework

### 3.1 Game Library Design

We develop a library of 13 matrix games spanning 7 core types from behavioral game theory, grouped into complete-information (fully known payoffs) and incomplete-information (need to reason with uncertainty) settings. Each game varies in stakes and strategic structure.

**Complete Information Games** include four classic structures as the examples in Table 1. Each cell displays the payoffs for Player 1 (row chooser) and Player 2 (column chooser), respectively. Players choose their strategies, and the payoffs reflect the resulting outcome for each pair of chosen actions.

**1. Competitive:** Each player's gain is the other's loss, which tests adversarial reasoning.
**2. Cooperation:** Risky coordination, where mutual cooperation yields the highest joint payoff.
**3. Mixed-Motive:** Social dilemmas balancing private vs. joint outcomes (e.g., Prisoner's Dilemma).
**4. Sequential:** temporal structure, with Player 1 moving first and Player 2 responding; we elicit decisions from the first mover.

Table 1: Representative matrices for complete information games

| (a) Competitive | | | (b) Cooperation | | (c) Mixed-Motive | | (d) Sequential | | |
|---|---|---|---|---|---|---|---|---|---|
| 10,-10 | 0,5 | -5,8 | 8,8 | 0,7 | 3,3 | 0,5 | 0,5 | 0,3 | 0,0 |
| -10,10 | 5,0 | 8,-5 | 7,0 | 5,5 | 5,0 | 1,1 | 5,2 | 3,3 | -1,-1 |
| 0,0 | 5,-5 | -5,5 | | | | | 2,4 | 4,3 | 0,-2 |

**Incomplete Information Games** incorporate belief-based reasoning about unknown payoff structures or player type. Examples of payoff matrices are shown in Table 2.

**1. Bayesian Coordination:** Players choose actions under uncertainty over the governing payoff matrix. Agents are given priors (e.g., 30% vs. 70%) over two types.
**2. Signaling:** The sender sees both real and fake matrices; the receiver must infer the true game. Payoffs reflect sender/receiver misalignment.

Table 2: Representative matrices for incomplete information games

| (a) Bayes type 1 | | (b) Bayes type 2 | | (c) Signaling (sender) | | (d) Signaling (receiver) | |
|---|---|---|---|---|---|---|---|
| 10,10 | 5,2 | 8,8 | 6,3 | 5,5 | 2,1 | 4,4 | 6,3 |
| 7,5 | 3,3 | 5,4 | 2,2 | 3,2 | 1,0 | 2,3 | 1,2 |

We also include the SW10 matrix from (54), a benchmark for identifying human reasoning levels. The complete payoff matrix tables are attached in the Appendix. While our experiments use abstract normal-form games $(N, \{A_i\}, \{u_i\})$, these payoffs directly mirror real-world interactions. For example, in a recommender system, the e-commerce platform chooses to recommend one of two items, $a_1$ or $a_2$, and the user choose to purchase ($b_1$) or not ($b_2$), yielding utilities $u_1(a_i, b_j)$ for the platform and $u_2(a_i, b_j)$ for the user. Similarly, in a negotiation, the proposer offers one of two deals, $a_1$ or $a_2$, and the responder either accepts ($b_1$) or rejects ($b_2$), capturing the payoffs of agreement. Thus, payoff matrices capture essential strategic structures in various decision-making scenarios.

## 3.2 Evaluation Procedure

To quantify the average strategic reasoning depth ($\tau$) from observed behaviors, we proceed as follows:

**Step 1. Empirical Choice Frequencies.** For each LLM and each game, we collect 30 independent trials and record the frequency $c_{ij}$ with which the model chose action $a_{ij}$.

**Step 2. Model-Implied Choice Probabilities.** Under the bounded-rationality framework, a level-$k$ agent's probability of selecting $a_{ij}$ is given by a logit function over its expected utility $U_{ij}^{(k)}$, scaled by $\lambda_k = \gamma \cdot k$. Aggregating across reasoning levels $k = 0, 1, 2, \ldots$ with Poisson weight $f_k(\tau)$ yields:

$$p_{ij}(\tau, \gamma) \;=\; \sum_{k=0} f_k(\tau) \, \frac{\exp(\gamma\, k\, U_{ij}^{(k)})}{\sum_{a \in A_i} \exp(\gamma\, k\, U_{ia}^{(k)})}, \quad f_k(\tau) = \frac{\tau^k e^{-\tau}}{k!}.$$

*Example:* In a two-player game where each player $i \in \{1, 2\}$ has actions $A_i = \{a_{i1}, a_{i2}\}$ and Player 1's utilities satisfy

$$u_1(a_{11}, a_{21}) = U_{11}, \quad u_1(a_{11}, a_{22}) = U_{12}, \quad u_1(a_{12}, a_{21}) = U_{21}, \quad u_1(a_{12}, a_{22}) = U_{22}$$

, one computes $U_{11}^{(k)}$ by summing each payoff $U_{1j}$ weighted by the probability that the opponent (a level-$(<k)$ mixture) plays the corresponding action. That expected utility then enters the probability function to yield $p_{11}^{(k)}$, and the mixture over $k$ produces $p_{11}(\tau, \gamma)$.

**Step 3. Maximum Likelihood Fit.** We estimate $(\tau, \gamma)$ by maximizing the log-likelihood of the observed counts under the model:

$$\max_{\tau, \gamma} \sum_{i,j} c_{ij} \, \ln p_{ij}(\tau, \gamma)$$

, where $c_{ij}$ is the count of times action $a_{ij}$ was chosen. Numerical optimization yields the values of $\tau$ that best fit the LLM's choice distribution.

## 3.3 Model Selection.

Given the rapid evolution of LLMs in both open- and closed-source communities, we select representative models from each to demonstrate our framework's effectiveness and highlight behavioral differences. While including more models is desirable, our selection captures common trends and supports meaningful evaluation. The selected models include GPT-4o (37), GPT-o1-preview (38), GPT-o3-mini (49), DeepSeek-R1 (27), DeepSeek-V2.5 (13), DeepSeek-V3 (14), gemma-2-27b-it (56), Gemini-2.0-Flash-Thinking (55), Granite-3.1-8B-Dense (47), Granite-3.1-3B-MoE, Claude-3-Opus (2), internlm2_5-20b-chat (6), Meta-LLaMA-3.1-405B-Instruct (26), Meta-LLaMA-3.1-8B-Instruct, QwQ-32B-Preview (57; 62), and glm-4-9b-chat (24).

## 3.4 Fit Diagnostics and Chance Baseline

We evaluate the maximum log-likelihood (MLL) for each model–game pair and contextualize these against a uniform **chance baseline**. For simultaneous games with $m$ possible actions for Player 1 and $n$ for Player 2, the baseline log-likelihood is defined as

$$\text{MLL}_{\text{chance}} = -\ln(mn).$$

For example,

$$2 \times 2 \Rightarrow \text{MLL}_{\text{chance}} = -\ln(4) = 1.386,$$
$$3 \times 3 \Rightarrow \text{MLL}_{\text{chance}} = -\ln(9) = 2.197.$$

For sequential games, since only the first mover's action is analyzed, the baseline reduces to

$$\text{MLL}_{\text{chance}} = -\ln(m),$$

where $m$ is the number of available actions (e.g., $m = 3 \Rightarrow 1.099$). Values close to this baseline indicate random or level-0–like behavior, whereas higher (less negative) MLL values reflect better fit quality. Appendix B reports per-model fits and baselines, complementing Table 3.

# 4 Evaluation Results and Comparison

In this section, we present reasoning depth evaluations, followed by an analysis of reasoning variation across game types through models' reasoning chains. By identifying distinct reasoning styles, we provide both quantitative and qualitative insights into how LLMs reason in interactive settings.

## 4.1 Strategic Reasoning Capabilities Across Models

Table 3 presents the reasoning depths evaluated through the framework for each model across games. Among all models, GPT-o1, GPT-o3-mini, and DeepSeek-R1 outperform others, with GPT-o1 ranking first in 6 games, GPT-o3-mini leading in 7 games, and DeepSeek-R1 securing the top position in 5 games. Some games have co-leaders, where multiple models achieve the highest evaluated strategic reasoning level. In these cases, certain LLMs consistently select strategies that align with theoretically robust principles while also demonstrating accurate anticipation of the opponent's behavior. However, not all models achieve this level of capability, indicating variations in reasoning depth. This shows both the strengths of top models in structured settings and the effectiveness of our framework in distinguishing optimal decision-making from suboptimal or inconsistent reasoning. Additionally, our findings reveal significant variations in reasoning capabilities across different game types.

GPT-o1 consistently ranks among the top models in competitive and incomplete-information games, indicating a strong capacity for goal-oriented and adversarial reasoning. DeepSeek-R1 excels in cooperative and mixed-motive settings, likely due to its reinforcement learning-based optimization for social interaction and mutual benefit. GPT-o3-mini performs robustly across all game types, particularly in cooperative, mixed-motive, and incomplete-information scenarios, reflecting a balanced ability to manage trade-offs, adapt to uncertainty, and apply probabilistic inference.

The performance gap between stronger models, such as DeepSeek-R1, GPT-o1 and GPT-o3-mini, and weaker ones, like DeepSeek V2.5, DeepSeek V3, and GPT-4o, is notable across most games. Nevertheless, advanced models do not always dominate all game types. Smaller models like Gemma-2-27B and LLaMA-3.1-8B sometimes match or outperform larger counterparts, and DeepSeek-R1 is a top model despite not being the largest. Additionally, derived from a much smaller base model, R1-distilled models can outperform significantly larger models [1] . This suggests that model size is not the sole factor in reasoning quality. Instead, context, model structure, and training data likely play a crucial role, highlighting the strong influence of contextual adaptation on LLMs' reasoning abilities.

Table 3: Strategic reasoning depth ($\tau$) across different game settings

| | Complete Information Game | | | | | | | | | | Incomplete Information Game | | |
| | Competitive Games | | | Cooperation Games | | | Mixed-motive Games | | | Seq. Games | Bayesian Games | | Signaling Games |
| | BL | HS | LS | BL | HP | AP | BL | H-Pun | L-Pun | | p=0.5 | p=0.9 | |
|---|---|---|---|---|---|---|---|---|---|---|---|---|---|
| GPT-4o | 1.543 | 1.936 | 0.729 | 0.602 | 0.143 | 1.505 | 1.665 | 0.537 | 1.499 | 0.222 | 1.973 | 1.953 | 3.590 |
| GPT-o1 | **4.741** | **4.311** | 3.126 | 2.800 | 0.628 | 1.322 | 0.143 | 0.069 | 1.481 | **2.846** | **4.225** | **4.225** | **3.980** |
| GPT-o3-mini | 1.309 | 2.087 | 2.430 | **3.550** | **4.226** | **3.944** | 3.352 | **3.944** | **2.931** | 0.635 | **4.225** | **4.225** | 3.079 |
| Gemma-V2 | 0.320 | 1.485 | 1.593 | 0.183 | 0.003 | 1.118 | 0.981 | 0.290 | 0.851 | 0.735 | 1.831 | 0.187 | 3.069 |
| Gemini-2.0 | 1.202 | 1.202 | 1.958 | 2.487 | 1.609 | 0.070 | 2.461 | 0.223 | 1.230 | 1.683 | **4.225** | **4.225** | 3.920 |
| Claude-3-Opus | 1.131 | 0.982 | 0.036 | 3.390 | 1.322 | 1.505 | 1.333 | 1.000 | 1.220 | 0.717 | 2.014 | **4.225** | 3.623 |
| Granite-3.1-8B | 1.428 | 0.003 | 3.429 | 1.127 | 0.128 | 1.476 | 1.360 | 0.198 | 1.376 | 1.480 | 1.191 | 1.261 | 1.527 |
| Granite-3.1-MoE | 1.098 | 1.257 | 1.336 | 3.396 | 2.405 | 2.515 | **3.895** | 3.524 | 2.508 | 1.560 | 2.119 | 2.407 | 2.832 |
| LLaMA-3.1-405B | 0.988 | 0.910 | 0.177 | 0.379 | 0.001 | 1.118 | 1.368 | 0.290 | 1.180 | 0.688 | 1.658 | 0.226 | 1.543 |
| LLaMA-3.1-8B | 1.298 | 0.964 | 1.548 | 1.066 | 0.717 | 0.000 | 1.431 | 0.173 | 1.475 | 0.357 | 0.340 | 0.069 | 2.336 |
| InternLM-V2 | 0.124 | 0.882 | 1.511 | 0.869 | 0.143 | 1.311 | 1.449 | 0.537 | 1.396 | 0.579 | 0.943 | 0.384 | 0.339 |
| QwQ-32B | 3.599 | 1.435 | **3.892** | 2.398 | 3.434 | 0.916 | 2.564 | 0.406 | 1.427 | 0.144 | 0.405 | 1.197 | 2.904 |
| GLM-4 | 1.177 | 0.962 | 1.488 | 1.250 | 0.436 | 1.247 | 1.392 | 0.173 | 1.481 | 0.748 | 0.810 | 0.639 | 1.252 |
| DeepSeek-V2.5 | 1.108 | 0.913 | 1.413 | 1.215 | 0.480 | 1.247 | 1.269 | 0.290 | 1.190 | 0.174 | 0.736 | 1.120 | 1.455 |
| DeepSeek-V3 | 0.142 | 0.390 | 0.042 | 0.183 | 0.003 | 1.118 | 1.399 | 0.173 | 1.346 | 0.883 | 1.295 | 1.347 | 3.410 |
| DeepSeek-R1 | 0.075 | 1.198 | 0.233 | **3.550** | **4.226** | **3.944** | 2.718 | 1.322 | 2.929 | 0.516 | **4.225** | **4.225** | 3.079 |

*Note:* **Bold values** indicate the highest strategic reasoning level ($\tau$) observed for that game. Abbreviations: HS = High Stake, LS = Low Stake, BL = Baseline, HP = High Payoff, AP = Asymmetric Payoff, H-Pun = High Punishment, L-Pun = Low Punishment.

## 4.2 Empirical Exploration for Variations of Leading LLM in Different Game Types

To investigate why certain advanced models perform well in specific game types while others do not, we focus our analysis on three top-performing models, DeepSeek-R1, GPT-o1, and GPT-o3-mini, in baseline competitive, cooperative, and mixed-motive games.

---

[1]Full results in Appendix B, Table 7

We gather available internal or summarized reasoning traces from each model and link them with its observed behavioral choices. For each model–game pair, we extract the empirical strategy distribution from 30 independent trials and identify the dominant or mixed-strategy pattern. We then examine the corresponding reasoning traces through qualitative analysis, aligning textual reasoning steps with revealed strategic behavior to evaluate coherence between what the model *says* and what it *does*. To ensure robustness and mitigate subjectivity, we collect a diverse set of reasoning traces from different models and anonymize them before analysis. Researchers who coded each CoT trace for reasoning style did so without knowing which model produced it, following a "blind review" procedure that minimizes post-hoc bias. A pre-defined structured coding rubric was used to classify reasoning styles (e.g., maximin, belief-based anticipation, opponent-oriented logic), and cross-validation was performed by having multiple researchers independently code overlapping subsets and compare results for consistency. Finally, we record the completion tokens required by each model to quantify computational demand and analyze whether longer reasoning chains correspond to improved or diminished decision quality.

Based on the summarization in Table 4, we observe that each model's performance in different game types reflects the strategic logic it tends to apply. While these styles are not mutually exclusive or deterministic, they offer a coherent lens for interpreting variation in performance.

Table 4: LLMs' reasoning styles and examples

| Model | Dominant Reasoning Style | Representative Excerpts |
|---|---|---|
| GPT-o1 | Consistent maximin, Payoff pessimism (risk-averse) | *"I select the row that maximizes our minimum payoff."* (competitive game) 
 *"Even though Row 0 gives 8 in the best case, Row 1 guarantees at least 5."* (coordinative game) 
 *"Worst-case payoff for me (assuming the opponent minimizes our payoff): 0"* (mixed strategy game) |
| GPT-o3 | Belief-guided decision-making with fallback to maximin | *"Player 2 is likely to choose column 1, so I respond with row 0."* (competitive game) 
 *"Column 2 guarantees the highest minimum payoff... That's why I selected column 2."* (mixed motive game) |
| DeepSeek-R1 | Opponent oriented decision, Lack of adversarial consideration | *"Player 2 will likely choose the column that maximizes their own payoff... so I choose row 0."* (competitive game) 
 *"... neither player benefits from deviating"* (coordinative game) |

GPT-o1 consistently applies minimax reasoning, evaluating each option by its worst-case outcome. In nearly every reasoning chain we extracted, the model explicitly states, "I will now use minimax to help me make a decision." This leads to strong performance in competitive games, where minimizing potential losses aligns with an optimal strategy. However, the same strategy becomes overly cautious in cooperative or mixed-motive games, where achieving mutual benefit often requires trust and risk-taking. GPT-o1 also exhibits a highly defensive stance, at times assuming that the opponent's objective is to minimize its payoff, interpreting interactions as fully adversarial. GPT-o3-mini demonstrates greater flexibility. It primarily engages in belief-based anticipation, attempting to infer the opponent's likely move and respond accordingly. This allows it to perform well in cooperative and mixed-motive settings, where alignment with the other player is key. Yet under uncertainty, GPT-o3-mini sometimes falls back to the worst-case logic, which can limit its potential in competitive games but also protects against major failures. DeepSeek-R1 adopts a strategy that begins with assumptions about the opponent's likely action, typically based on the idea that the opponent will pursue their own payoff. This logic works well in cooperative games, where the players' incentives are aligned, but lacks the adversarial caution required in competitive games. R1 also tends to exhibit strategic trust, assuming that when mutual benefit is unavailable, the opponent will not deviate simply to harm the other player, in contrast to GPT-o1, which at times assumes spiteful intent.

An additional observation relates to token usage, as shown in Appendix B Table 8. Contrary to intuitive expectations, higher token counts in internal reasoning do not correlate with better reasoning. In fact, GPT-o1 and DeepSeek-R1 are leaders in competitive and cooperative games, respectively, while they produce the shortest CoT outputs within their strongest games. We observe that longer reasoning chains often suggest hesitation and uncertainty, rather than deeper insight. Conversely, more concise CoT responses tend to signal higher confidence and more direct strategy choices. Notably, in competitive games, DeepSeek-R1 often exhibits repeated self-doubt in its CoT, but this results in redundant reasoning loops that inflate token usage without improvement. This points to future research on improving token efficiency in CoT generation, such as incorporating RL-based training that encourages decisive and diverse reasoning over repetitive patterns.

Table 5: CoT effect on reasoning examples

| Model | Prompt Excerpt |
|---|---|
| **Example of Improvement:** 

 Claude-3-opus | *"I should pick the row that gives me the maximum value among these worst-case...* 
 *Wait, I made a mistake... Let me reconsider using the concept of maximin..."* 
 *"Row 0 and Row 2 both have a worst-case of –5, but Row 0 offers a potential payoff of 10,* 
 *so I choose Row 0."* |
| **Example of Decline:** 
 GPT-4o | *"We assume Player One picks the row that minimizes our payoff, so..."* 
 *"Break tie with average... Column 2 is better."* |

## 4.3 Effect of CoT Prompting

Our empirical study of top-performing models with embedded internal CoT reasoning motivates us to explore whether CoT prompting can also enhance models that do not natively generate internal reasoning. The strategic reasoning level after explicitly adding the CoT prompt is presented in Appendix B Table 9. Intuitively, many believe CoT can likely enhance a model's reasoning ability. However, our results show that CoT's impact on strategic reasoning is mixed, with no consistent improvement across all models and game types, which is consistent with prior findings suggesting that the "CoT does not always help" (18; 10), particularly in strategic settings.

For example, Figure 2 compares $\tau$ in the Competitive-Base game with and without CoT prompting. While some models improve, others remain unchanged or even decline. We select two representative cases: with the largest improvement and sharpest decline: Claude-3-Opus and GPT-4o. Table 5 presents excerpts from their CoT reasoning chains to illustrate the underlying dynamics [2].

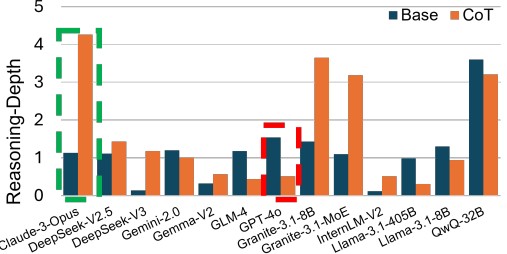

Figure 2: Comparison between baseline and CoT
*Note*: Green marks the largest increase; red marks the largest decline

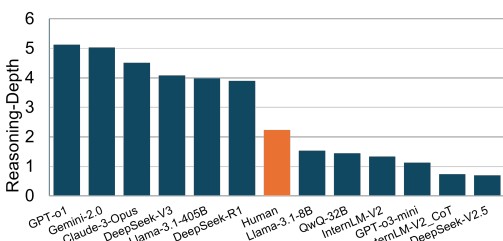

Figure 3: Comparison among human participants and LLMs on the SW game

If we look at their gaming strategies, Claude-3-Opus quickly identifies a clear attempt to apply the maximin strategy at an earlier stage, an appropriate approach for the competitive game. It initially makes calculation errors, referring to incorrect cells during its payoff analysis, but the model explicitly identifies and corrects these mistakes mid-chain. This self-correction enables it to recover and ultimately arrive at a robust decision that aligns with its strategic intent. In contrast, GPT-4o starts with a fundamentally flawed assumption, mistakenly believing that the opponent will act adversarially to minimize its payoff. Based on this incorrect premise, the model applies a tie-breaking rule using average values across columns, stuck in a verbose reasoning loop that reinforces the wrong strategy. These again imply that CoT does not guarantee better reasoning. When a model begins with a sound strategy, CoT can support useful self-correction and refinement. However, when the underlying reasoning is misaligned from the beginning, extended chains often lead to repetitive and unproductive loops rather than meaningful improvement, which echoes one of our findings in Section 4.2.

## 5 Detecting Social Effect with Human

While reasoning depth is useful for evaluating LLMs' strategic capacity, model sophistication alone is not enough. Real-world deployment requires alignment with contextual goals, ethical constraints, and human norms. To examine these factors, we compare LLM reasoning with human performance in identical games and examine how assigned demographic personas influence model behavior, shedding light on identity-induced strategic shifts and fairness in AI decision-making.

---

[2]Full reasoning chains are in Appendix E

Figure 3 compares all models against human subjects in a single-scenario setting (54; 51), with six models surpassing human performance. However, this ranking should not imply that "a higher score is better". In many applications, alignment with human reasoning is more critical than outperformance. Our framework supports such evaluation by quantifying reasoning depth across subjects, offering a path to calibrate LLMs for human-aligned behavior. For instance, businesses can use representative human samples to fine-tune models toward user-aligned decisions.

To further evaluate the alignment of LLMs with human strategic reasoning, we construct a diverse set of socio-demographic personas spanning a broad range of individual characteristics, inspired by (29). These personas are structured into two tiers: foundational demographic features (Panel 1) and advanced identity attributes (Panel 2), as detailed in Table 6.

Table 6: The Personas across 10 socio-demographic groups that we explore in this study.

| Group | Persona |
|---|---|
| **Panel 1: Foundational Demographic Features** | |
| Sex | male, female |
| Education Level | below lower secondary, lower secondary, upper secondary, short-cycle tertiary, bachelor, and graduate degrees |
| Marital Status | never married, married, widowed, divorced |
| Living Area | rural, urban |
| Age | 15 - 24, 25 - 34, 35 - 44, 45 - 54, 55 - 64, 65+ |
| **Panel 2: Advanced Demographic Features** | |
| Sex Orientation | heterosexual, homosexual, bisexual, asexual |
| Disability | physically-disabled, able-bodied |
| Race | African, Hispanic, Asian, Caucasian |
| Religion | Jewish, Christian, Atheist, Other Religious |
| Political Affiliation | lifelong Democrat, lifelong Republican, Barack Obama supporter, Donald Trump supporter |

We then perform Ordinary Least Squares (OLS) regressions using the estimated behavioral parameters as dependent variables. The general specification of the regression is:

$$\text{Reasoning\_Depth}_i = \beta_0 + \beta_n D_{ni} + \epsilon_i$$

Here, $i$ indexes each observation, $D_{ni}$ is the $n$-th binary demographic indicator, $\beta_n$ denotes its estimated effect on the behavioral parameter, and $\epsilon_i$ is the error term.

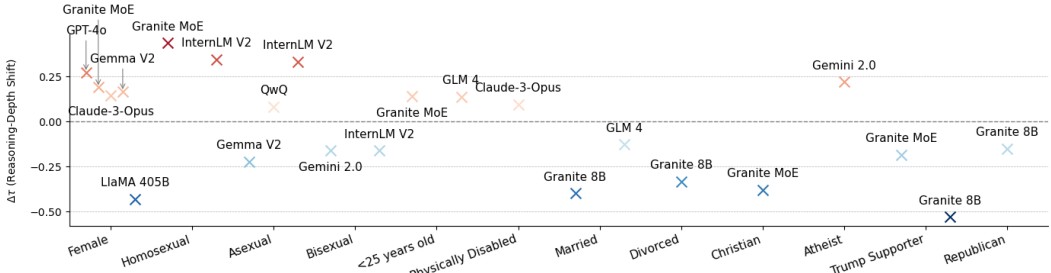

Figure 4: Significant demographic shifts in reasoning depth

*Note:* Each point reflects the significant change in reasoning depth relative to a reference group through OLS. Reference categories are: Male (gender), Heterosexual (sexual orientation), Age 25–64 (age), Able-bodied (disability status), Single (marital status), Other Religious (religion), and Democrat (political affiliation). Positive (negative) shifts indicate higher (lower) reasoning depth relative to the reference group.

Figure 4 shows estimated shifts in reasoning depth across demographic conditions with statistically significant effects from OLS regressions. A consistent pattern emerges around gender: when prompted with a "female" persona, several large models—such as GPT-4o, InternLM V2, and Claude 3-Opus—exhibit increased reasoning depth, suggesting deeper multi-level reasoning under female identity framing. Sexuality-based variation reveals further heterogeneity. While "homosexual" and "bisexual" personas raise reasoning depth in models like GPT-4o and GLM 4, they lead to significantly reduced $\tau$ in Gemini 2.0 and Granite MoE. These divergent effects highlight both the contextual flexibility of LLM reasoning and the risk of identity-induced attenuation. Other demographic factors can also cause disturbance in reasoning depths. We include full results of this study in Appendix B.

**Mechanism note.** Observed persona-induced shifts likely stem from statistical associations in training corpora that link demographic descriptors to behavioral patterns, potentially modulated by reinforcement learning from human feedback (RLHF). Models often deny explicit bias yet reveal

implicit adaptation under identical payoffs, indicating latent contextual priors. Our framework surfaces these implicit shifts and quantifies their magnitude for downstream auditing and fairness alignment.

# 6    Conclusion and Discussion

This study sets a new milestone in evaluating LLM strategic reasoning by assessing while accounting for contexts, addressing the limitations of NE-focused benchmarks. We find that strong performance on standard tasks (e.g., math) does not always guarantee higher reasoning depth. Analysis also reveals the limitations of CoT prompting in strategic settings, where its benefits are not universal and can, at times, impair reasoning. These findings inform ongoing efforts to refine LLM reasoning workflows for complex decision-making. Additionally, our demographic analysis exposes how biases can emerge even in top-performing models, underscoring the need for fairness-aware evaluation.

**Effectiveness of CoT.** While CoT can enhance the reasoning process in language models, its effectiveness hinges on task comprehension and alignment with intended objectives. Our analysis shows that well-aligned CoT prompts support structured reasoning and improved outcomes, whereas misaligned prompts can amplify errors—even in models trained with RL from Human Feedback (RLHF). DeepSeek-R1 illustrates that, despite strong performance in some games, it underperforms in others, exhibiting repetitive self-questioning and redundant token generation without better solutions. RLHF helps refine strategies, but without the intrinsic ability to evaluate context-dependent strategies, models may remain trapped in ineffective reasoning loops. These findings highlight the need for adaptive, context-aware reasoning and raise a key open question: how can external prompts be better aligned with a model's internal reasoning processes to ensure robust and interpretable behavior?

**Alignment and Fairness in AI Reasoning.** Superior strategic reasoning depth does not equate to ethical decision-making. When demographic cues are introduced, models can adjust their strategies based on the given role, indicating that their decisions can encode and present societal biases. This is especially concerning in multi-agent interactions, where solely prioritizing strategic efficiency may lead to inequitable or adversarial outcomes. In interactive settings, LLMs that adjust strategies based on demographic cues risk perpetuating bias in negotiations and resource allocation, leading to real-world inequalities over time. This presents a critical challenge to balance strategic reasoning with fairness. Unconstrained optimization may encourage exploitative tactics, while rigid fairness constraints could hinder adaptability. Future development must move beyond performance metrics to integrate ethical safeguards, ensuring that LLMs engage in fair and justifiable decision-making in dynamic multi-agent environments.

**Limitations and Future Directions:** While our study sheds light on LLM reasoning patterns and benchmark alignment, the causal link between prompts and decision outcomes remains underexplored. Future work should investigate how individual cognitive skills—such as logic, math, and contextual understanding—interact to shape reasoning capabilities. Moreover, our current framework is semi-dynamic; extending it to real-time multi-agent interactions may reveal new reasoning dynamics and performance disparities. Exploring such settings could provide deeper insights into adaptive LLM behavior and strategy formation.

**Broader Impact:** This study provides practical insights into how the internal decision-making processes of large language models translate into real-world social outcomes through quantitative evaluations. Unlike explicit discrimination, these biases emerge through adaptive reasoning, making them more difficult to detect and regulate. As organizations increasingly rely on AI to guide complex decisions affecting diverse populations, researchers and practitioners must remain cautious: subtle reasoning biases in sophisticated AI could inadvertently widen existing societal gaps. Addressing these challenges today will ensure that LLMs become inclusive tools promoting societal equity, rather than unintentionally reinforcing historical injustices.

# Acknowledgment

We gratefully thank the anonymous reviewers and area chair for their insightful comments and constructive feedback. We also acknowledge the support provided in part by the AMD Center of Excellence at the University of Illinois Urbana–Champaign.

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

# A Detailed Model Specification

In this appendix we provide a fully detailed statement of the model and then work through a concrete two-player, two-action example for clarity.

## A.1 Game and Payoff Structure

- $\mathcal{I} = \{1, \ldots, N\}$: set of players.
- $A_i$: finite action set of player $i$.
- $A_{-i} = \prod_{j \neq i} A_j$: joint action space of all players except $i$.
- $a_i \in A_i$, $a_{-i} \in A_{-i}$: player $i$'s action and opponents' profile.
- $u_i(a_i, a_{-i})$: payoff to $i$ when playing $a_i$ against $a_{-i}$.

## A.2 Reasoning Levels and Noise

- Agents draw a *reasoning level* $k \in \{0, 1, 2, \ldots\}$ from a Poisson distribution with mean $\tau > 0$:

$$f_k \;=\; \Pr(\text{level} = k) = \frac{\tau^k e^{-\tau}}{k!}, \quad k = 0, 1, 2, \ldots$$

- Each level-$k$ agent has *precision* (inverse noise)

$$\lambda_k = \gamma\, k, \quad \gamma > 0.$$

Higher $\lambda_k$ means more sharply payoff-driven (less random) choices.

## A.3 Level-$h$ Strategy Distributions

For each opponent $j \neq i$ and each potential level $h$, we assume a *level-h strategy*

$$\sigma_j^{(h)} : A_j \to [0, 1], \quad \sum_{a_j \in A_j} \sigma_j^{(h)}(a_j) = 1.$$

This is the probability that a level-$h$ thinker $j$ chooses action $a_j$.

## A.4 Mixed Marginal Belief

A level-$k$ agent does not know which of the levels $0, \ldots, k-1$ its opponents occupy, so it forms a *mixture* of their level-$h$ strategies. Define the normalized weight

$$W_h^{(k)} = \frac{f_h}{\sum_{\ell=0}^{k-1} f_\ell}, \quad h = 0, \ldots, k-1.$$

Then the agent's marginal belief about opponent $j$ is

$$\bar{\sigma}_j^{(k)}(a_j) = \sum_{h=0}^{k-1} W_h^{(k)}\, \sigma_j^{(h)}(a_j), \quad a_j \in A_j.$$

## A.5 Joint Belief

The notation

$$a_{-i} = (a_1, \ldots, a_{i-1}, a_{i+1}, \ldots, a_N)$$

stands for the actions of all players except $i$. For example, if $N = 3$ and $i = 1$, then $a_{-1} = (a_2, a_3)$, meaning "opponent 2 plays $a_2$" and "opponent 3 plays $a_3$." Under the assumption that each opponent $j \neq i$ acts independently, in this example, $\Pr(a_2, a_3) = \bar{\sigma}_2^{(k)}(a_2)\, \bar{\sigma}_3^{(k)}(a_3)$, In general, the probability of the profile $a_{-i}$ is

$$\Pr(a_{-i}) \;=\; \prod_{j \neq i} \bar{\sigma}_j^{(k)}(a_j).$$

### A.6 Expected Utility at Level $k$

Given belief $\Pr(a_{-i})$, the expected payoff for player $i$ choosing $a_{ij} \in A_i$ is

$$U_{ij}^{(k)} = \sum_{a_{-i} \in A_{-i}} \Pr(a_{-i})\, u_i\big(a_{ij}, a_{-i}\big).$$

### A.7 Choice Probabilities

Once a level-$k$ agent has computed the expected payoffs $U_{ij}^{(k)}$ for each action $a_{ij}$, it converts these into a probability distribution via a logit rule. The precision parameter $\lambda_k$ governs how sharply the agent favors utility-oriented actions:

$$p_{ij}^{(k)} = \frac{\exp\big(\lambda_k\, U_{ij}^{(k)}\big)}{\sum_{a \in A_i} \exp\big(\lambda_k\, U_{ia}^{(k)}\big)}.$$

When $\lambda_k$ is small, payoffs have little influence and choices are nearly uniform; as $\lambda_k \to \infty$, the agent approaches deterministic best-response behavior.

### A.8 Population-Level Aggregation

Because agents' reasoning levels $k$ are drawn from the Poisson weights $f_k$, the overall probability that a randomly chosen agent plays action $a_{ij}$ is the weighted average of level-specific probabilities:

$$p_{ij} = \sum_{k=0}^{\infty} f_k\, p_{ij}^{(k)}.$$

This mixture captures heterogeneity in both depth of reasoning and choice stochasticity across the population. In implementation, the infinite sum is truncated at a sufficiently large $K$ so that $\sum_{k=0}^{K} f_k \approx 1$.

# B Additional Results

In this section, we present the complete results for the following model variants across all game settings: vanilla models, vanilla models with CoT prompting, distilled models based on DeepSeek-R1, vanilla models with embedded demographic features, and vanilla models with both CoT prompting and demographic features.

## Distilled Model

We evaluate DeepSeek-R1 distilled models across multiple architectures—LLaMA-70B, LLaMA-8B, Qwen-32B, Qwen-14B, Qwen-7B, and Qwen-1.5B—to assess whether the distillation process introduces additional biases in the inherited models. As shown in Table 7, R1-distilled models can outperform significantly larger counterparts, despite originating from smaller base models. This indicates that model size alone does not determine reasoning quality. Instead, factors such as architectural design, training data, and contextual adaptation play a pivotal role in shaping LLM reasoning capabilities.

However, our analysis reveals that knowledge distillation can also introduce new biases, as evidenced in Table 11b. While distillation enhances reasoning performance—particularly in cooperative and mixed-motive game settings—it also increases sensitivity to demographic features. In the vanilla LLaMA models, even with CoT prompting, bias remains limited, typically affecting only one or two attributes. In contrast, R1-distilled versions of LLaMA-8B and LLaMA-70B exhibit amplified demographic biases. For instance, DSk-R1-LLaMA70B shows marked sensitivity to age and race.

These findings highlight the critical importance of monitoring the distillation process. If the teacher model is trained on malicious or skewed data, these biases may propagate to the student model, potentially resulting in harmful behaviors. More concerningly, even when trained on ostensibly neutral data, hidden biases can be transferred during distillation. This underscores the need for rigorous auditing and bias mitigation when applying knowledge distillation in LLM development.

## CoT + Demographic Features

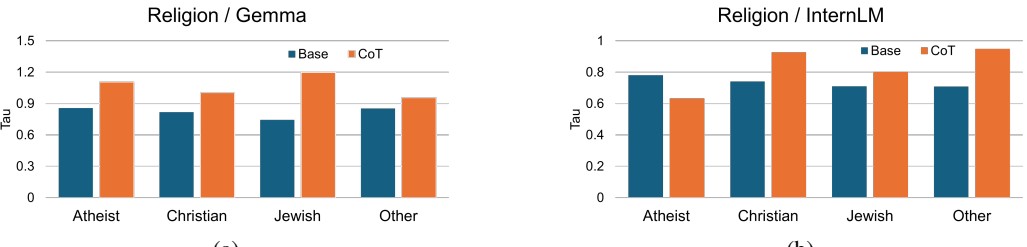

Figure 5: Impact of CoT on the same models with the same demographic features

To assess whether explicit step-by-step reasoning affects model fairness, we introduce CoT prompting and analyze its impact on demographic correlations, as reported in Table 12. Overall, we observe that CoT increases the number of demographic variables exhibiting significant influence—even in models that were previously neutral—indicating that CoT can shift how models interpret and weigh user cues. For instance, Figure 5 shows that Gemma-2 exhibits a marked increase in sensitivity to the "Jewish" attribute, while InternLM, initially unbiased, develops religious disparities under CoT. These findings raise important fairness concerns: CoT can both surface latent biases and introduce new correlations, making fairness properties highly sensitive to prompt design. In particular, if a model overemphasizes certain demographic signals, CoT may exacerbate rather than alleviate bias. This highlights the necessity of prompt-specific bias audits and careful monitoring as CoT prompting becomes more widely adopted.

Table 7: Strategic Reasoning Performance ($\tau$) Across Different Game Settings Include Distilled Models

| | Complete Information Game | | | | | | | | | Incomplete Information Game | | | |
| | Competitive Games | | | Cooperation Games | | | Mixed-motive Games | | | Seq. Games | Bayesian Games | | Signaling Games |
| | BL | HS | LS | BL | HP | AP | BL | H-Pun | L-Pun | | p=0.5 | p=0.9 | |
|---|---|---|---|---|---|---|---|---|---|---|---|---|---|
| GPT-4o | 1.543 | 1.936 | 0.729 | 0.602 | 0.143 | 1.505 | 1.665 | 0.537 | 1.499 | 0.222 | 1.973 | 1.953 | 3.590 |
| GPT-o1 | **4.741** | **4.311** | 3.126 | 2.800 | 0.628 | 1.322 | 0.143 | 0.069 | 1.481 | **2.846** | 4.225 | **4.225** | 3.980 |
| GPT-o3-mini | 1.309 | 2.087 | 2.430 | **3.550** | **4.226** | **3.944** | 3.352 | **3.944** | 2.931 | 0.635 | **4.225** | **4.225** | 3.079 |
| Gemma-V2 | 0.320 | 1.485 | 1.593 | 0.183 | 0.003 | 1.118 | 0.981 | 0.290 | 0.851 | 0.735 | 1.831 | 0.187 | 3.069 |
| Gemini-2.0 | 1.202 | 1.202 | 1.958 | 2.487 | 1.609 | 0.070 | 2.461 | 0.223 | 1.230 | 1.683 | **4.225** | **4.225** | 3.920 |
| Claude-3-Opus | 1.131 | 0.982 | 0.036 | 3.390 | 1.322 | 1.505 | 1.333 | 1.000 | 1.220 | 0.717 | 2.014 | **4.225** | 3.623 |
| Granite-3.1-8B | 1.428 | 0.003 | 3.429 | 1.127 | 0.128 | 1.476 | 1.360 | 0.198 | 1.376 | 1.480 | 1.191 | 1.261 | 1.527 |
| Granite-3.1-MoE | 1.098 | 1.257 | 1.336 | 3.396 | 2.405 | 2.515 | **3.895** | 3.524 | 2.508 | 1.560 | 2.119 | 2.407 | 2.832 |
| LLaMA-3.1-405B | 0.988 | 0.910 | 0.177 | 0.379 | 0.001 | 1.118 | 1.368 | 0.290 | 1.180 | 0.688 | 1.658 | 0.226 | 1.543 |
| LLaMA-3.1-8B | 1.298 | 0.964 | 1.548 | 1.066 | 0.717 | 0.000 | 1.431 | 0.173 | 1.475 | 0.357 | 0.340 | 0.069 | 2.336 |
| InternLM-V2 | 0.124 | 0.882 | 1.511 | 0.869 | 0.143 | 1.311 | 1.449 | 0.537 | 1.396 | 0.579 | 0.943 | 0.339 | 0.339 |
| QwQ-32B | 3.599 | 1.435 | **3.892** | 2.398 | 3.434 | 0.916 | 2.564 | 0.406 | 1.427 | 0.144 | 0.405 | 1.197 | 2.904 |
| GLM-4 | 1.177 | 0.962 | 1.488 | 1.250 | 0.436 | 1.247 | 1.392 | 0.173 | 1.481 | 0.748 | 0.810 | 0.639 | 1.252 |
| DeepSeek-V2.5 | 1.108 | 0.913 | 1.413 | 1.215 | 0.480 | 1.247 | 1.269 | 0.290 | 1.190 | 0.174 | 0.736 | 1.120 | 1.455 |
| DeepSeek-V3 | 0.142 | 0.390 | 0.042 | 0.183 | 0.003 | 1.118 | 1.399 | 0.173 | 1.346 | 0.883 | 1.295 | 1.347 | 3.410 |
| DeepSeek-R1 | 0.075 | 1.198 | 0.233 | **3.550** | **4.226** | **3.944** | 2.718 | 1.322 | 2.929 | 0.516 | **4.225** | **4.225** | 3.079 |
| **DeepSeek-R1-Distill Models:** | | | | | | | | | | | | | |
| LLaMA-70B | 0.488 | 0.028 | 0.072 | 0.730 | 0.619 | 1.512 | 2.376 | 0.745 | 1.228 | 0.884 | **4.225** | 3.066 | 3.460 |
| LLaMA-8B | 0.341 | 0.261 | 0.150 | 0.628 | - | 0.000 | 1.394 | 0.633 | 1.266 | 0.068 | 1.796 | 1.966 | 3.245 |
| Qwen-1.5B | 0.421 | 0.434 | 0.013 | 2.180 | 0.990 | 0.916 | 1.420 | 0.397 | 1.224 | 0.119 | 0.883 | 0.884 | 1.430 |
| Qwen-14B | 0.518 | 0.633 | 0.047 | 2.487 | 0.606 | 1.945 | 1.445 | 1.179 | 2.033 | 0.097 | 1.980 | 1.407 | **3.791** |
| Qwen-32B | 0.790 | 1.045 | 0.335 | 1.093 | 0.916 | 0.406 | 2.472 | 0.773 | 1.399 | 0.075 | **4.225** | **4.225** | 3.731 |
| Qwen-7B | 0.120 | 1.045 | 0.084 | 0.907 | 0.163 | 1.470 | 1.457 | 0.677 | 1.147 | 1.528 | 0.645 | 0.994 | 3.112 |

*Note:* **Bold values** indicate the highest strategic reasoning level ($\tau$) observed for that game. A dash (-) indicates cases where the model struggles to establish a stable level of reasoning due to potential convergence challenges. Abbreviations: **HS** = High Stake, **LS** = Low Stake, **BL** = Baseline, **HP** = High Payoff, **AP** = Asymmetric Payoff, **H-Pun** = High Punishment, **L-Pun** = Low Punishment.

Table 8: Summary Statistics of Leading Models' Completion Token Numbers for Baseline Games

| | Competitive games | | | Cooperative games | | | Mixed-strategy games | | |
| | GPT-o1 | GPT-o3-mini | DeepSeek-R1 | GPT-o1 | GPT-o3-mini | DeepSeek-R1 | GPT-o1 | GPT-o3-mini | DeepSeek-R1 |
|---|---|---|---|---|---|---|---|---|---|
| **mean** | 7747.27 | 8938.33 | 10979.93 | 3197.80 | 5966.80 | 1764.67 | 3472.87 | 5944.07 | 2318.33 |
| **min** | 3994 | 6200 | 9050 | 2431 | 4786 | 1001 | 2585 | 4586 | 1599 |
| **max** | 14858 | 13223 | 13350 | 5690 | 7100 | 2931 | 5020 | 7659 | 3742 |

Table 9: Strategic Reasoning Performance ($\tau$) Across Different Game Settings in CoT

| | Complete Information Game | | | | | | | | | Incomplete Information Game | | | |
| | Competitive Games | | | Cooperation Games | | | Mixed-motive Games | | | Seq. Games | Bayesian Games | | Signaling Games |
| | BL | HS | LS | BL | HP | AP | BL | H-Pun | L-Pun | | p=0.5 | p=0.9 | |
|---|---|---|---|---|---|---|---|---|---|---|---|---|---|
| Claude-3-Opus-CoT | **4.264** | 3.327 | - | 0.270 | 1.053 | 1.311 | 1.181 | 0.127 | 0.140 | 1.692 | 0.502 | 0.564 | 1.489 |
| DeepSeek-V2.5-CoT | 1.427 | 1.867 | 1.103 | 3.046 | 2.510 | 1.376 | 2.608 | 1.322 | 3.319 | 1.646 | 1.058 | 0.854 | 0.405 |
| DeepSeek-V3-CoT | 1.179 | 0.480 | - | 3.269 | 1.322 | 0.628 | 2.815 | 1.253 | 2.831 | 0.369 | 0.913 | 1.039 | 0.079 |
| Gemma-V2-CoT | 0.566 | 0.848 | 1.469 | 0.629 | 0.986 | 0.070 | 0.511 | 0.069 | 0.916 | 0.717 | 1.202 | 0.782 | 1.419 |
| Gemini-2.0-CoT | 1.005 | 1.410 | - | 1.080 | 0.777 | 1.515 | 1.350 | 0.896 | 1.314 | 0.209 | 0.879 | 1.057 | 1.529 |
| GPT-4o-CoT | 0.511 | 0.288 | 1.063 | 0.871 | 0.762 | 1.515 | 1.411 | 0.629 | 1.311 | 1.596 | **4.225** | 3.066 | 3.976 |
| Granite-3.1-8B-CoT | 3.646 | 1.048 | **4.247** | 1.990 | 1.022 | 0.333 | 1.426 | 0.020 | 1.480 | 1.440 | 2.141 | 1.138 | 0.908 |
| Granite-3.1-3B MoE-CoT | 3.193 | 3.301 | 0.159 | 0.400 | 0.177 | **2.256** | 1.518 | 1.160 | 1.139 | **2.843** | 1.956 | 0.223 | 2.009 |
| LLaMA-3.1-405B-CoT | 0.314 | 1.416 | 0.049 | 0.357 | 0.835 | 0.560 | 0.154 | 0.223 | 0.693 | 0.182 | 0.738 | 0.857 | 1.521 |
| LLaMA-3.1-8B-CoT | 0.942 | 1.007 | 0.263 | 0.118 | 0.829 | 0.342 | 1.383 | 0.960 | 0.158 | 0.152 | 0.736 | 0.981 | 1.511 |
| InternLM-V2-CoT | 0.512 | 1.039 | 1.552 | 1.067 | 1.021 | 1.247 | 1.371 | 0.927 | 1.364 | 0.077 | 1.068 | 1.220 | 2.338 |
| QwQ-32B-CoT | 3.207 | 1.491 | **3.956** | **3.550** | **4.226** | 0.628 | **3.352** | **3.944** | 3.080 | 0.068 | 0.438 | 1.024 | **4.226** |
| GLM-4-CoT | 0.442 | 0.645 | 0.221 | 0.069 | 1.119 | 0.406 | 0.069 | 0.795 | 0.223 | 0.360 | 1.058 | 0.871 | 1.486 |

*Note:* **Bold values** indicate the highest strategic reasoning level ($\tau$) observed for that game. A dash (-) indicates cases where the model struggles to establish a stable level of reasoning due to potential convergence challenges. Abbreviations: **HS** = High Stake, **LS** = Low Stake, **BL** = Baseline, **HP** = High Payoff, **AP** = Asymmetric Payoff, **H-Pun** = High Punishment, **L-Pun** = Low Punishment.

Table 10: Context Controller ($\gamma$) Across Different Game Settings

| | Complete Information Game | | | | | | | | | Incomplete Information Game | | | |
| | Competitive Games | | | Cooperation Games | | | Mixed-motive Games | | | Seq. Games | Bayesian Games | | Signaling Games |
| | BL | HS | LS | BL | HP | AP | BL | H-Pun | L-Pun | | p=0.5 | p=0.9 | |
|---|---|---|---|---|---|---|---|---|---|---|---|---|---|
| GPT-4o | 1.486 | 0.595 | 4.076 | 0.000 | 0.000 | 5.000 | 0.846 | 0.950 | 0.236 | 0.000 | 0.000 | 0.000 | 34.198 |
| GPT-o1 | 0.064 | 0.029 | 1.231 | 5.000 | 0.047 | 5.000 | 5.001 | 5.276 | 0.395 | 10.152 | 1.000 | 0.000 | 0.676 |
| GPT-o3-mini | 1.482 | 0.595 | 1.316 | 5.000 | 0.041 | 5.000 | 5.001 | 5.276 | 0.120 | 0.089 | 5.000 | 0.071 | 35.734 |
| Gemma-V2 | 22.898 | 0.592 | 3.010 | 0.000 | 0.000 | 1.045 | 0.031 | 0.317 | 0.129 | 0.000 | 0.000 | 0.000 | 52.362 |
| Gemini-2.0 | 1.471 | 0.568 | 3.119 | 1.004 | 0.023 | 5.000 | 5.001 | 5.276 | 0.300 | 0.044 | 1.000 | 0.000 | 0.000 |
| Claude-3-Opus | 0.000 | 0.003 | 17.497 | 0.000 | 0.061 | 5.000 | 5.000 | 5.276 | 0.355 | 0.000 | 0.000 | 0.000 | 38.675 |
| Granite-3.1-8B | 0.125 | 0.000 | 0.144 | 0.000 | 0.000 | 5.000 | 0.000 | 0.000 | 0.000 | 0.000 | 5.000 | 0.000 | 0.000 |
| Granite-3.1-MoE | 2.383 | 0.931 | 3.874 | 5.000 | 0.040 | 5.000 | 5.000 | 1.010 | 2.860 | 0.065 | 5.000 | 0.054 | 0.000 |
| LLaMA-3.1-405B | 1.461 | 0.589 | 16.412 | 0.000 | 0.000 | 5.000 | 0.025 | 0.441 | 0.000 | 0.000 | 0.000 | 0.000 | 37.102 |
| LLaMA-3.1-8B | 0.000 | 0.000 | 0.000 | 1.009 | 0.000 | 5.000 | 0.337 | 5.861 | 0.081 | 0.000 | 0.000 | 0.000 | 48.384 |
| InternLM-V2 | 16.153 | 0.000 | 0.000 | 0.000 | 0.000 | 5.000 | 0.000 | 0.593 | 0.061 | 0.000 | 0.000 | 0.000 | 42.306 |
| QwQ-32B | 0.045 | 0.921 | 0.143 | 5.000 | 0.032 | 0.056 | 6.609 | 0.842 | 0.882 | 0.109 | 5.000 | 0.331 | 48.972 |
| GLM-4 | 0.000 | 0.000 | 0.000 | 0.000 | 0.000 | 0.000 | 0.000 | 0.000 | 0.000 | 0.000 | 0.000 | 0.000 | 38.796 |
| DeepSeek-V2.5 | 0.000 | 0.000 | 0.000 | 0.000 | 0.000 | 0.000 | 0.000 | 0.000 | 0.000 | 0.000 | 0.000 | 0.000 | 41.697 |
| DeepSeek-V3 | 0.706 | 0.569 | 21.732 | 5.000 | 0.041 | 5.000 | 5.001 | 5.276 | 0.120 | 0.054 | 5.000 | 0.078 | 52.656 |
| DeepSeek-R1 | 13.719 | 0.583 | 19.847 | 0.000 | 0.000 | 1.045 | 0.000 | 0.014 | 0.156 | 0.000 | 0.000 | 0.000 | 40.615 |

*Note:* Abbreviations: HS = High Stake, LS = Low Stake, BL = Baseline, HP = High Payoff, AP = Asymmetric Payoff, H-Pun = High Punishment, L-Pun = Low Punishment.

Table 11: LLMs Sensitivity to Demographic Features through Regression Analysis

(a) SOTA LLM models

| Feature | GPT 4o | GPT o1 | GPT o3 | Granite 3.1 MoE | Granite 3.1 8B | Claude 3 Opus | Gemma V2 | Gemini 2.0 | LLaMA 405B | LLaMA 8B | InternLM V2 | QwQ | GLM 4 |
|---|---|---|---|---|---|---|---|---|---|---|---|---|---|
| <25 years old | -0.296 | -0.051 | 0.020 | **0.140** | -0.017 | -0.071 | -0.106 | -0.090 | 0.177 | -0.001 | 0.026 | 0.027 | **0.138** |
| | (0.184) | (0.260) | (0.130) | **(0.079)** | (0.207) | (0.067) | (0.140) | (0.084) | (0.338) | (0.088) | (0.175) | (0.090) | **(0.076)** |
| >55 years old | 0.094 | -0.226 | 0.036 | -0.098 | 0.246 | -0.009 | 0.083 | 0.054 | 0.190 | 0.030 | -0.055 | -0.101 | 0.027 |
| | (0.138) | (0.197) | (0.098) | (0.060) | (0.156) | (0.051) | (0.106) | (0.063) | (0.255) | (0.066) | (0.132) | (0.068) | (0.058) |
| Female | **0.274*** | 0.263 | 0.118 | **0.190*** | 0.153 | **0.146**** | **0.167** | -0.057 | **-0.432** | 0.063 | 0.073 | 0.067 | -0.047 |
| | **(0.126)** | (0.183) | (0.091) | **(0.056)** | (0.145) | **(0.047)** | **(0.098)** | (0.059) | **(0.237)** | (0.062) | (0.123) | (0.063) | (0.053) |
| Graduate Level | 0.208 | -0.283 | -0.029 | -0.070 | 0.255 | -0.069 | 0.020 | 0.053 | 0.131 | 0.034 | -0.076 | -0.122 | **0.141*** |
| | (0.170) | (0.245) | (0.122) | (0.075) | (0.195) | (0.063) | (0.132) | (0.078) | (0.317) | (0.083) | (0.165) | (0.082) | **(0.069)** |
| Below Secondary | **0.275*** | -0.154 | -0.011 | 0.095 | 0.128 | -0.077 | 0.030 | -0.090 | 0.323 | -0.064 | -0.120 | 0.006 | -0.004 |
| | **(0.139)** | (0.196) | (0.098) | (0.060) | (0.156) | (0.050) | (0.106) | (0.063) | (0.254) | (0.066) | (0.132) | (0.066) | (0.056) |
| Divorced | 0.062 | 0.041 | -0.014 | -0.049 | **-0.336** | -0.024 | -0.179 | -0.066 | 0.211 | -0.032 | 0.166 | 0.050 | 0.023 |
| | (0.163) | (0.237) | (0.118) | (0.072) | **(0.188)** | (0.061) | (0.128) | (0.076) | (0.307) | (0.080) | (0.159) | (0.078) | (0.066) |
| Married | 0.039 | -0.340 | -0.155 | -0.057 | **-0.401*** | -0.111 | -0.177 | -0.125 | 0.166 | 0.004 | 0.156 | -0.032 | **-0.129*** |
| | (0.166) | (0.239) | (0.119) | (0.073) | **(0.190)** | (0.061) | (0.129) | (0.077) | (0.310) | (0.081) | (0.161) | (0.077) | **(0.065)** |
| Widowed | 0.097 | -0.081 | -0.034 | 0.043 | -0.321 | -0.016 | 0.024 | -0.063 | 0.016 | 0.022 | 0.203 | -0.019 | 0.059 |
| | (0.175) | (0.246) | (0.122) | (0.075) | (0.195) | (0.063) | (0.132) | (0.079) | (0.319) | (0.083) | (0.165) | (0.078) | (0.066) |
| Rural | 0.076 | -0.285 | 0.086 | -0.028 | -0.009 | -0.001 | 0.099 | -0.036 | -0.289 | 0.048 | -0.011 | 0.037 | 0.052 |
| | (0.126) | (0.179) | (0.089) | (0.055) | (0.142) | (0.046) | (0.096) | (0.057) | (0.232) | (0.060) | (0.120) | (0.058) | (0.049) |
| Asexual | -0.178 | 0.198 | -0.066 | -0.100 | 0.042 | 0.076 | **-0.224** | -0.022 | 0.230 | -0.010 | **0.333*** | **0.082** | 0.016 |
| | (0.171) | (0.248) | (0.123) | (0.076) | (0.197) | (0.064) | **(0.134)** | (0.080) | (0.321) | (0.084) | **(0.167)** | **(0.080)** | (0.067) |
| Bisexual | -0.176 | 0.342 | 0.055 | -0.122 | -0.004 | 0.025 | -0.099 | **-0.162** | -0.254 | -0.145 | **0.438** | -0.075 | -0.017 |
| | (0.190) | (0.277) | (0.138) | (0.084) | (0.220) | (0.071) | (0.149) | **(0.089)** | (0.359) | (0.094) | **(0.186)** | (0.092) | (0.077) |
| Homosexual | -0.262 | -0.111 | -0.076 | -0.002 | **-0.418*** | 0.086 | -0.160 | -0.057 | -0.044 | -0.143 | **0.345** | 0.109 | -0.035 |
| | (0.183) | (0.263) | (0.131) | (0.080) | **(0.209)** | (0.068) | (0.142) | (0.084) | (0.341) | (0.089) | **(0.177)** | (0.086) | (0.072) |
| Physically Disabled | -0.169 | -0.144 | 0.047 | 0.036 | 0.045 | **0.096*** | -0.099 | -0.033 | -0.151 | 0.095 | 0.012 | 0.000 | -0.049 |
| | (0.125) | (0.181) | (0.090) | (0.055) | (0.144) | **(0.047)** | (0.098) | (0.058) | (0.235) | (0.061) | (0.122) | (0.062) | (0.053) |
| African | 0.033 | 0.302 | 0.035 | 0.080 | -0.197 | -0.063 | -0.105 | 0.037 | -0.207 | 0.013 | -0.079 | -0.036 | 0.052 |
| | (0.158) | (0.226) | (0.113) | (0.069) | (0.180) | (0.058) | (0.122) | (0.072) | (0.293) | (0.076) | (0.152) | (0.074) | (0.062) |
| Asian | 0.267 | 0.086 | -0.081 | 0.102 | -0.001 | -0.015 | -0.187 | 0.115 | -0.160 | 0.126 | 0.061 | 0.082 | 0.083 |
| | (0.182) | (0.263) | (0.131) | (0.080) | (0.209) | (0.068) | (0.142) | (0.084) | (0.341) | (0.089) | (0.177) | (0.091) | (0.076) |
| Hispanic | 0.129 | 0.098 | -0.070 | 0.007 | -0.123 | -0.026 | 0.071 | -0.043 | -0.046 | 0.026 | -0.134 | -0.112 | 0.022 |
| | (0.158) | (0.224) | (0.112) | (0.068) | (0.178) | (0.058) | (0.121) | (0.072) | (0.291) | (0.076) | (0.151) | (0.075) | (0.063) |
| Atheist | -0.132 | 0.000 | -0.034 | -0.086 | -0.265 | -0.036 | -0.017 | **0.220**** | -0.075 | -0.139 | 0.038 | 0.024 | 0.026 |
| | (0.183) | (0.259) | (0.129) | (0.079) | (0.206) | (0.065) | (0.140) | **(0.083)** | (0.336) | (0.088) | (0.175) | (0.090) | (0.076) |
| Christian | 0.407 | 0.172 | -0.014 | -0.042 | **-0.381** | -0.004 | -0.019 | 0.062 | -0.357 | -0.052 | -0.059 | -0.069 | -0.022 |
| | (0.169) | (0.246) | (0.123) | (0.075) | **(0.196)** | (0.063) | (0.133) | (0.079) | (0.319) | (0.083) | (0.166) | (0.085) | (0.072) |
| Jewish | -0.008 | 0.251 | 0.043 | -0.075 | -0.176 | 0.001 | -0.144 | -0.070 | -0.149 | -0.107 | -0.009 | -0.028 | -0.042 |
| | (0.149) | (0.215) | (0.107) | (0.066) | (0.171) | (0.055) | (0.116) | (0.069) | (0.279) | (0.073) | (0.145) | (0.075) | (0.063) |
| Obama Supporter | -0.098 | -0.381 | -0.022 | -0.086 | -0.073 | -0.068 | -0.026 | 0.048 | -0.091 | -0.022 | -0.086 | -0.042 | -0.096 |
| | (0.171) | (0.243) | (0.121) | (0.074) | (0.194) | (0.063) | (0.131) | (0.078) | (0.316) | (0.082) | (0.164) | (0.079) | (0.066) |
| Trump Supporter | -0.028 | 0.012 | 0.049 | **-0.187*** | **-0.533*** | 0.016 | 0.116 | 0.022 | -0.576 | -0.110 | -0.175 | 0.040 | -0.034 |
| | (0.179) | (0.258) | (0.128) | **(0.079)** | **(0.205)** | (0.066) | (0.139) | (0.083) | (0.334) | (0.087) | (0.174) | (0.082) | (0.069) |
| Republican | -0.035 | -0.106 | 0.068 | **-0.152*** | -0.223 | -0.030 | 0.104 | -0.055 | -0.299 | -0.032 | -0.206 | -0.010 | 0.005 |
| | (0.165) | (0.234) | (0.116) | **(0.071)** | (0.186) | (0.060) | (0.126) | (0.075) | (0.303) | (0.079) | (0.157) | (0.076) | (0.064) |
| Constant | 0.431 | 1.304 | 0.813 | 1.501 | 2.048 | 0.170 | 0.945 | 1.412 | 2.079 | 1.084 | 0.527 | 0.427 | 0.961 |
| | (0.262) | (0.377) | (0.188) | (0.115) | (0.300) | (0.097) | (0.203) | (0.121) | (0.489) | (0.127) | (0.254) | (0.028) | (0.022) |

(Continued) LLMs Sensitivity to Demographic Features through Regression Analysis

(b) DeepSeek family models

| Feature | DS V2.5 | DS V3 | DS R1 | DSk-R1 Qwen 1.5B | DSk-R1 Qwen 7B | DSk-R1 Qwen 14B | DSk-R1 Qwen 32B | DSk-R1 LLaMA 8B | DSk-R1 LLaMA 70B |
|---|---|---|---|---|---|---|---|---|---|
| <25 years old | -0.047 | 0.002 | 0.154 | 0.224 | 0.180 | 0.170 | 0.017 | -0.041 | -0.098 |
|  | (0.036) | (0.036) | (0.109) | (0.270) | (0.177) | (0.114) | (0.256) | (0.094) | (0.127) |
| >55 years old | -0.038 | -0.014 | 0.032 | 0.092 | **-0.235** | -0.003 | 0.042 | -0.039 | **-0.242*** |
|  | (0.028) | (0.027) | (0.082) | (0.204) | **0.133** | (0.086) | (0.194) | (0.071) | **0.095** |
| Female | 0.023 | -0.033 | 0.027 | -0.245 | 0.167 | -0.003 | 0.115 | -0.086 | -0.043 |
|  | (0.026) | (0.025) | (0.076) | (0.189) | (0.124) | (0.080) | (0.180) | (0.066) | (0.089) |
| Graduate Level | 0.005 | -0.006 | -0.026 | -0.417 | 0.016 | **0.201** | 0.139 | -0.137 | -0.048 |
|  | (0.034) | (0.034) | (0.102) | (0.254) | (0.166) | **0.107** | (0.241) | (0.088) | (0.119) |
| Below Secondary | 0.016 | 0.016 | **-0.179*** | **-0.371** | 0.063 | -0.029 | 0.272 | -0.026 | 0.133 |
|  | (0.027) | (0.027) | **0.082** | **0.203** | (0.133) | (0.086) | (0.193) | (0.071) | (0.095) |
| Divorced | **-0.070*** | -0.052 | -0.120 | -0.157 | **0.531**** | -0.144 | 0.277 | 0.046 | 0.066 |
|  | **0.033** | (0.033) | (0.099) | (0.245) | **0.161** | (0.104) | (0.233) | (0.085) | (0.115) |
| Married | **-0.057** | 0.013 | -0.102 | 0.273 | 0.152 | -0.125 | 0.035 | 0.113 | -0.106 |
|  | **0.033** | (0.033) | (0.100) | (0.248) | (0.162) | (0.105) | (0.235) | (0.086) | (0.116) |
| Widowed | **-0.059** | 0.004 | -0.040 | 0.046 | 0.060 | -0.242 | -0.170 | 0.029 | 0.050 |
|  | **0.034** | (0.034) | (0.103) | (0.255) | (0.167) | (0.108) | (0.242) | (0.088) | (0.119) |
| Rural | -0.014 | -0.019 | **0.157*** | -0.144 | -0.029 | -0.037 | -0.095 | -0.005 | -0.065 |
|  | (0.025) | (0.025) | **0.075** | (0.185) | (0.121) | (0.078) | (0.176) | (0.064) | (0.087) |
| Asexual | 0.038 | 0.018 | 0.016 | 0.164 | 0.070 | -0.081 | -0.094 | -0.007 | 0.167 |
|  | (0.035) | (0.034) | (0.104) | (0.257) | (0.168) | (0.109) | (0.244) | (0.089) | (0.120) |
| Bisexual | **0.078*** | **0.081*** | 0.109 | 0.338 | -0.109 | 0.095 | **-0.545*** | -0.071 | 0.109 |
|  | **0.039** | **0.038** | (0.116) | (0.287) | (0.188) | (0.122) | **0.273** | (0.100) | (0.135) |
| Homosexual | 0.029 | 0.012 | **0.303**** | 0.147 | -0.054 | -0.145 | -0.132 | **-0.163** | 0.141 |
|  | (0.037) | (0.037) | **0.110** | (0.273) | (0.179) | (0.115) | (0.259) | **0.095** | (0.128) |
| Physically Disabled | -0.024 | -0.003 | 0.081 | 0.138 | 0.089 | -0.073 | 0.062 | 0.020 | 0.051 |
|  | (0.025) | (0.025) | (0.076) | (0.188) | (0.123) | (0.080) | (0.178) | (0.065) | (0.088) |
| African | -0.037 | **0.067*** | 0.079 | 0.071 | -0.152 | -0.073 | -0.094 | 0.070 | -0.098 |
|  | (0.032) | **0.031** | (0.094) | (0.234) | (0.153) | (0.099) | (0.222) | (0.081) | (0.110) |
| Asian | -0.011 | 0.044 | 0.050 | -0.207 | **-0.359*** | -0.106 | 0.073 | 0.074 | **-0.298*** |
|  | (0.037) | (0.036) | (0.110) | (0.273) | **0.179** | (0.115) | (0.259) | (0.095) | **0.128** |
| Hispanic | 0.005 | 0.044 | -0.038 | -0.107 | **-0.302*** | -0.033 | 0.190 | 0.069 | -0.170 |
|  | (0.031) | (0.031) | (0.094) | (0.233) | **0.152** | (0.098) | (0.221) | (0.081) | (0.109) |
| Atheist | 0.017 | -0.001 | -0.036 | 0.255 | -0.026 | -0.077 | **0.445** | 0.005 | 0.108 |
|  | (0.036) | (0.036) | (0.108) | (0.269) | (0.176) | (0.114) | **0.255** | (0.093) | (0.126) |
| Christian | -0.022 | -0.028 | 0.005 | -0.051 | **-0.364*** | -0.003 | 0.227 | 0.142 | 0.012 |
|  | (0.034) | (0.034) | (0.103) | (0.255) | **0.167** | (0.108) | (0.242) | (0.089) | (0.120) |
| Jewish | -0.013 | 0.024 | 0.037 | 0.234 | -0.174 | 0.110 | -0.036 | 0.087 | 0.021 |
|  | (0.030) | (0.030) | (0.090) | (0.223) | (0.146) | (0.094) | (0.212) | (0.077) | (0.105) |
| Obama Supporter | -0.010 | **-0.068*** | 0.102 | 0.130 | -0.142 | -0.021 | -0.191 | -0.131 | 0.081 |
|  | (0.034) | **0.034** | (0.102) | (0.252) | (0.165) | (0.107) | (0.239) | (0.088) | (0.118) |
| Trump Supporter | -0.008 | -0.051 | 0.070 | 0.023 | -0.072 | 0.066 | -0.373 | -0.034 | 0.236 |
|  | (0.036) | (0.036) | (0.108) | (0.267) | (0.175) | (0.113) | (0.254) | (0.093) | (0.125) |
| Republican | -0.016 | -0.022 | 0.075 | 0.083 | -0.153 | 0.093 | **-0.456*** | -0.014 | 0.078 |
|  | (0.033) | (0.032) | (0.098) | (0.242) | (0.159) | (0.103) | **0.230** | (0.084) | (0.114) |
| Constant | 0.934 | 0.373 | 0.125 | 1.456 | 0.891 | 0.518 | 0.852 | 1.242 | 0.391 |
|  | (0.053) | (0.052) | (0.158) | (0.391) | (0.256) | (0.166) | (0.371) | (0.136) | (0.183) |

*Note*: The values in each cell represent the estimated coefficient, with the standard error provided in parentheses below. Bolded coefficients indicate statistical significance at the 90% confidence level. The asterisks (*, **, ***) denote the levels of statistical significance as follows: *: $p < 0.05$ (significant at the 95% confidence level), **: $p < 0.01$ (significant at the 99% confidence level), ***: $p < 0.001$ (significant at the 99.9% confidence level).

Table 12: LLMs Sensitivity to Demographic Features with Chain-of-Thought through Regression Analysis

| Feature | GPT 4o | Granite 3.1 MoE | Granite 3.1 8B | Claude 3 Opus | DS V2.5 | DS V3 | Gemma V2 | Gemini 2.0 | LLaMA 8B | LLaMA 405B | InternLM V2 | QwQ | GLM 4 |
|---|---|---|---|---|---|---|---|---|---|---|---|---|---|
| <25 years old | -0.003 | -0.085 | 0.525 | 0.022 | 0.108 | 0.046 | -0.010 | -0.007 | -0.086 | -0.022 | **0.388** | -0.002 | -0.114 |
|  | (0.042) | (0.323) | (0.407) | (0.097) | (0.451) | (0.051) | (0.136) | (0.100) | (0.241) | (0.128) | **(0.225)** | (0.106) | (0.158) |
| >55 years old | 0.036 | **0.640**\*\* | 0.087 | -0.056 | 0.079 | **0.065** | 0.010 | 0.057 | -0.040 | 0.078 | 0.193 | 0.102 | -0.035 |
|  | (0.031) | **(0.244)** | (0.307) | (0.073) | (0.340) | **(0.039)** | (0.103) | (0.075) | (0.182) | (0.096) | (0.170) | (0.080) | (0.119) |
| Female | -0.036 | 0.279 | 0.013 | -0.056 | **-0.700**\* | 0.027 | -0.122 | 0.067 | -0.191 | 0.005 | 0.056 | **0.217**\*\* | 0.173 |
|  | (0.029) | (0.227) | (0.285) | (0.068) | **(0.316)** | (0.036) | (0.095) | (0.070) | (0.169) | (0.089) | (0.158) | **(0.074)** | (0.111) |
| Graduate Level | -0.014 | 0.328 | 0.332 | -0.032 | 0.067 | **-0.084** | -0.108 | 0.086 | -0.225 | 0.074 | 0.093 | -0.066 | -0.047 |
|  | (0.040) | (0.304) | (0.382) | (0.091) | (0.423) | **(0.048)** | (0.128) | (0.094) | (0.227) | (0.120) | (0.211) | (0.100) | (0.148) |
| Below Secondary | -0.003 | -0.106 | 0.156 | 0.025 | -0.106 | -0.064 | -0.040 | 0.015 | 0.070 | -0.085 | 0.153 | -0.085 | 0.066 |
|  | (0.032) | (0.243) | (0.306) | (0.073) | (0.339) | (0.039) | (0.102) | (0.075) | (0.182) | (0.096) | (0.169) | (0.080) | (0.119) |
| Divorced | -0.043 | 0.208 | **0.657** | -0.053 | 0.621 | -0.011 | **-0.219** | 0.068 | -0.056 | 0.129 | 0.161 | 0.018 | 0.095 |
|  | (0.037) | (0.294) | **(0.369)** | (0.088) | (0.410) | (0.047) | **(0.124)** | (0.091) | (0.219) | (0.116) | (0.204) | (0.096) | (0.143) |
| Married | -0.018 | -0.168 | -0.077 | **0.179**\* | 0.225 | 0.001 | -0.036 | **-0.186**\* | 0.024 | 0.100 | 0.071 | -0.032 | 0.092 |
|  | (0.038) | (0.297) | (0.373) | **(0.089)** | (0.414) | (0.047) | (0.125) | **(0.092)** | (0.221) | (0.117) | (0.206) | (0.097) | (0.145) |
| Widowed | **-0.067** | -0.123 | 0.204 | 0.030 | 0.407 | -0.051 | **-0.308**\* | -0.060 | 0.348 | 0.163 | -0.276 | -0.066 | 0.106 |
|  | **(0.040)** | (0.305) | (0.383) | (0.091) | (0.425) | (0.048) | **(0.128)** | (0.094) | (0.228) | (0.120) | (0.212) | (0.100) | (0.149) |
| Rural | -0.001 | -0.305 | 0.010 | 0.077 | -0.190 | 0.012 | 0.030 | -0.066 | -0.066 | -0.078 | **-0.258** | 0.101 | 0.002 |
|  | (0.029) | (0.222) | (0.279) | (0.067) | (0.309) | (0.035) | (0.093) | (0.069) | (0.166) | (0.088) | **(0.154)** | (0.073) | (0.108) |
| Asexual | -0.061 | **-0.574** | 0.294 | 0.117 | -0.104 | 0.057 | -0.140 | 0.070 | 0.064 | -0.016 | -0.039 | **0.179** | -0.069 |
|  | (0.040) | **(0.308)** | (0.387) | (0.092) | (0.429) | (0.049) | (0.129) | (0.095) | (0.230) | (0.121) | (0.214) | **(0.101)** | (0.150) |
| Bisexual | -0.035 | **-0.936**\*\* | 0.620 | 0.101 | -0.080 | -0.023 | -0.181 | -0.134 | -0.105 | 0.085 | 0.305 | -0.144 | 0.161 |
|  | (0.043) | **(0.344)** | (0.432) | (0.103) | (0.479) | (0.055) | (0.145) | (0.106) | (0.257) | (0.136) | (0.239) | (0.113) | (0.168) |
| Homosexual | -0.003 | -0.230 | 0.458 | 0.134 | -0.171 | 0.024 | -0.184 | 0.037 | 0.285 | 0.098 | 0.142 | -0.073 | -0.010 |
|  | (0.042) | (0.327) | (0.411) | (0.098) | (0.455) | (0.052) | (0.137) | (0.101) | (0.244) | (0.129) | (0.227) | (0.107) | (0.159) |
| Physically Disabled | -0.010 | 0.323 | 0.190 | -0.036 | 0.312 | 0.019 | -0.040 | 0.033 | -0.033 | 0.107 | 0.054 | -0.018 | -0.168 |
|  | (0.029) | (0.225) | (0.283) | (0.068) | (0.314) | (0.036) | (0.095) | (0.070) | (0.168) | (0.089) | (0.157) | (0.074) | (0.110) |
| African | -0.044 | 0.123 | -0.558 | -0.038 | **0.802**\* | -0.016 | 0.135 | -0.009 | 0.017 | 0.113 | -0.104 | 0.062 | -0.133 |
|  | (0.037) | (0.280) | (0.353) | (0.084) | **(0.391)** | (0.044) | (0.118) | (0.087) | (0.209) | (0.111) | (0.195) | (0.092) | (0.137) |
| Asian | -0.048 | **0.859**\*\* | **-1.107**\*\* | -0.034 | 0.341 | -0.023 | 0.212 | -0.071 | 0.109 | 0.183 | -0.114 | 0.127 | -0.244 |
|  | (0.043) | **(0.326)** | **(0.410)** | (0.098) | (0.455) | (0.052) | (0.137) | (0.101) | (0.244) | (0.129) | (0.227) | (0.107) | (0.159) |
| Hispanic | -0.034 | -0.151 | -0.442 | -0.018 | -0.001 | -0.012 | 0.051 | 0.027 | 0.059 | 0.038 | 0.028 | 0.023 | -0.049 |
|  | (0.036) | (0.279) | (0.350) | (0.084) | (0.388) | (0.044) | (0.117) | (0.086) | (0.208) | (0.110) | (0.194) | (0.091) | (0.136) |
| Atheist | -0.017 | 0.303 | 0.204 | -0.024 | -0.114 | -0.026 | **0.233** | 0.142 | 0.338 | -0.041 | **-0.472** | -0.071 | **0.616**\*\*\* |
|  | (0.042) | (0.322) | (0.405) | (0.097) | (0.449) | (0.051) | **(0.136)** | (0.099) | (0.240) | (0.127) | **(0.224)** | (0.106) | **(0.157)** |
| Christian | -0.051 | 0.310 | 0.510 | -0.033 | -0.251 | -0.004 | 0.091 | 0.041 | 0.191 | -0.003 | -0.139 | -0.075 | 0.161 |
|  | (0.038) | (0.306) | (0.384) | (0.092) | (0.426) | (0.048) | (0.129) | (0.094) | (0.228) | (0.121) | (0.213) | (0.100) | (0.149) |
| Jewish | -0.004 | 0.067 | 0.534 | 0.015 | 0.393 | **-0.073** | **0.316**\*\* | -0.051 | **0.388** | 0.000 | -0.139 | -0.011 | 0.095 |
|  | (0.035) | (0.267) | (0.336) | (0.080) | (0.372) | **(0.042)** | **(0.112)** | (0.082) | **(0.199)** | (0.105) | (0.186) | (0.088) | (0.130) |
| Obama Supporter | 0.025 | -0.341 | -0.102 | 0.146 | -0.255 | -0.004 | -0.160 | 0.178 | 0.289 | -0.021 | -0.022 | -0.033 | -0.051 |
|  | (0.039) | (0.302) | (0.380) | (0.091) | (0.421) | (0.048) | (0.127) | (0.093) | (0.225) | (0.119) | (0.210) | (0.099) | (0.147) |
| Trump Supporter | -0.026 | -0.141 | 0.274 | -0.085 | **-0.851** | 0.054 | **-0.413**\*\* | 0.191 | -0.132 | 0.189 | 0.079 | -0.068 | -0.121 |
|  | (0.041) | (0.320) | (0.402) | (0.096) | **(0.446)** | (0.051) | **(0.135)** | (0.099) | (0.239) | (0.126) | (0.223) | (0.105) | (0.156) |
| Republican | 0.019 | 0.232 | 0.108 | -0.088 | -0.212 | -0.007 | -0.142 | -0.031 | -0.284 | 0.179 | 0.036 | -0.138 | -0.073 |
|  | (0.038) | (0.290) | (0.365) | (0.087) | (0.404) | (0.046) | (0.122) | (0.090) | (0.217) | (0.114) | (0.202) | (0.095) | (0.142) |
| Constant | 4.183 | 2.382 | 1.742 | 0.247 | 1.911 | 0.328 | 1.363 | 1.245 | 0.490 | 0.187 | 0.809 | 0.374 | 0.423 |
|  | (0.058) | (0.468) | (0.589) | (0.140) | (0.653) | (0.074) | (0.197) | (0.145) | (0.350) | (0.185) | (0.326) | (0.154) | (0.229) |

*Note*: The values in each cell represent the estimated coefficient, with the standard error provided in parentheses below. Bolded coefficients indicate statistical significance at the 90% confidence level. The asterisks (\*, \*\*, \*\*\*) denote the levels of statistical significance as follows: \*: $p < 0.05$ (significant at the 95% confidence level), \*\*: $p < 0.01$ (significant at the 99% confidence level), \*\*\*: $p < 0.001$ (significant at the 99.9% confidence level).

Table 13: Likelihood Across Different Game Settings

| | Complete Information Game | | | | | | | | | Incomplete Information Game | | | |
| | Competitive Games | | | Cooperation Games | | | Mixed-motive Games | | | Seq. Games | Bayesian Games | | Signaling Games |
| | BL | HS | LS | BL | HP | AP | BL | H-Pun | L-Pun | | p=0.5 | p=0.9 | |
|---|---|---|---|---|---|---|---|---|---|---|---|---|---|
| GPT-4o | -1.498 | -1.363 | -1.932 | -1.386 | -1.386 | -1.368 | -0.635 | -0.646 | -1.129 | -0.693 | -1.386 | -1.386 | -1.045 |
| GPT-o1 | -1.849 | -2.003 | -1.020 | -1.086 | -1.304 | -1.160 | -0.023 | -0.023 | -0.801 | -1.368 | -1.384 | -1.386 | -0.742 |
| GPT-o3-mini | -1.427 | -1.066 | -1.169 | -0.708 | -1.253 | -0.023 | -0.023 | -0.023 | -1.336 | -1.256 | -0.708 | -1.332 | -0.780 |
| Gemma-V2 | -2.045 | -1.484 | -1.818 | -1.386 | -1.386 | -1.386 | -1.375 | -1.379 | -1.326 | -1.386 | -1.386 | -1.386 | -0.675 |
| Gemini-2.0 | -1.603 | -1.663 | -1.623 | -1.384 | -1.386 | -0.650 | -0.023 | -0.023 | -0.965 | -1.365 | -1.366 | -1.386 | -1.099 |
| Claude-3-Opus | -2.197 | -2.197 | -2.194 | -1.386 | -1.177 | -0.785 | -0.490 | -0.023 | -0.934 | -1.386 | -1.386 | -1.386 | -0.657 |
| Granite-3.1-8B | -2.164 | -2.197 | -2.135 | -1.386 | -1.386 | -1.372 | -1.386 | -1.386 | -1.386 | -1.386 | -1.370 | -1.386 | -2.197 |
| Granite-3.1-MoE | -1.945 | -1.897 | -1.901 | -0.861 | -1.280 | -0.367 | -0.454 | -0.478 | -0.303 | -1.162 | -0.752 | -1.364 | -2.197 |
| LLaMA-3.1-405B | -1.670 | -1.819 | -2.154 | -1.386 | -1.386 | -1.386 | -1.381 | -1.373 | -1.386 | -1.386 | -1.386 | -1.386 | -0.693 |
| LLaMA-3.1-8B | -2.197 | -2.197 | -2.197 | -1.386 | -1.386 | -1.386 | -1.367 | -1.382 | -1.373 | -1.386 | -1.386 | -1.386 | -0.996 |
| InternLM-V2 | -2.164 | -2.197 | -2.197 | -1.386 | -1.386 | -1.368 | -1.386 | -1.339 | -1.386 | -1.386 | -1.386 | -1.386 | -0.903 |
| QwQ-32B | -2.162 | -1.884 | -2.104 | -1.194 | -1.359 | -0.841 | -1.273 | -0.987 | -0.638 | -1.281 | -1.330 | -1.216 | -1.080 |
| GLM-4 | -2.197 | -2.197 | -2.197 | -1.386 | -1.386 | -1.386 | -1.386 | -1.386 | -1.386 | -1.386 | -1.386 | -1.386 | -0.871 |
| DeepSeek-V2.5 | -2.197 | -2.197 | -2.197 | -1.386 | -1.386 | -1.386 | -1.386 | -1.386 | -1.386 | -1.386 | -1.386 | -1.386 | -1.078 |
| DeepSeek-V3 | -2.195 | -1.683 | -2.104 | -0.708 | -1.253 | -0.023 | -0.023 | -0.023 | -1.336 | -1.349 | -1.086 | -1.323 | -0.876 |
| DeepSeek-R1 | -2.148 | -2.111 | -2.193 | -1.386 | -1.386 | -1.386 | -1.386 | -1.385 | -1.277 | -1.386 | -1.386 | -1.386 | -0.758 |

*Note:* Abbreviations: HS = High Stake, LS = Low Stake, BL = Baseline, HP = High Payoff, AP = Asymmetric Payoff, H-Pun = High Punishment, L-Pun = Low Punishment.

# C   Game Library Design

In this manuscript, we have developed and collected multiple games, the payoff matrices are below. The complete information includes:

### Table 14: Competitive Games

(a) Base

| | | |
|---|---|---|
| 10, -10 | 0, 5 | -5, 8 |
| -10, 10 | 5, 0 | 8, -5 |
| 0, 0 | 5, -5 | -5, 5 |

(b) High-stake

| | | |
|---|---|---|
| 20, -20 | 0, 10 | -10, 15 |
| -20, 20 | 10, 0 | 15, -10 |
| 0, 0 | 10, -10 | -10, 10 |

(c) Low-stake

| | | |
|---|---|---|
| 3, -3 | 0, 1 | -1, 2 |
| -3, 3 | 1, 0 | 2, -1 |
| 0, 0 | 1, -1 | -1, 1 |

### Table 15: Cooperation Games (Stag Hunt)

(a) Base

| | |
|---|---|
| 8, 8 | 0, 7 |
| 7, 0 | 5, 5 |

(b) High-payoff

| | |
|---|---|
| 20, 20 | 0, 7 |
| 7, 0 | 5, 5 |

(c) Asymmetric-payoff

| | |
|---|---|
| 12, 8 | 0, 7 |
| 7, 0 | 5, 5 |

### Table 16: Mixed-Motive Games (Prisoner's Dilemma)

(a) Base

| | |
|---|---|
| 3, 3 | 0, 5 |
| 5, 0 | 1, 1 |

(b) High-punishment

| | |
|---|---|
| 10, 10 | 0, 15 |
| 15, 0 | -5, 5 |

(c) Low-punishment

| | |
|---|---|
| 3, 3 | 0, 4 |
| 4, 0 | 2, 2 |

1. **Competitive Games**: These games model adversarial interactions where one player's gain is exactly balanced by the other player's loss. The game consists of three variations:
   - Baseline: A standard competitive game where players choose strategies to maximize their own payoff while minimizing their opponent's.
   - High-Stake: A version where payoff differences are significantly larger, increasing the risk and reward of each decision.
   - Low-Stake: A variant where payoff differences are minimized, reducing the overall impact of each decision and testing an agent's ability to navigate lower-risk scenarios.

2. **Cooperation Games (Stag Hunt)**: These games examine the trade-off between individual risk and collective reward, highlighting the challenge of trust and cooperation. Three variations are included:
   - Baseline: The classic Stag Hunt game, where mutual cooperation leads to the highest payoff, but unilateral deviation results in significant losses.
   - High-Payoff: A modified version where the rewards for successful cooperation are increased, testing whether agents are more willing to take cooperative risks.
   - Asymmetric-Payoff: A variant where one player receives a higher payoff for cooperation than the other, introducing an imbalance that challenges fairness and trust dynamics.

3. **Mixed-Motive Games (Prisoner's Dilemma)**: These games explore the tension between cooperation and self-interest, where defection offers short-term individual benefits but harms collective outcomes. Variations include:
   - Baseline: The standard Prisoner's Dilemma setup, where mutual cooperation yields moderate rewards, but unilateral defection offers a higher individual payoff at the expense of the other player.
   - High-Punishment: A version where the penalty for defection is significantly increased, discouraging selfish behavior.
   - Low-Punishment: A variant where the cost of defection is minimal, encouraging more frequent defection and testing an agent's ability to recognize long-term cooperation benefits.

4. **Sequential Games**: These games test strategic planning and reaction-based decision-making by introducing turn-based interactions.
   - Player 1 makes the first move, setting the stage for Player 2's decision.

Table 17: Sequential Games

| | | |
|---|---|---|
| 0, 5 | 0, 3 | 0, 0 |
| 5, 2 | 3, 3 | -1, -1 |
| 2, 4 | 4, 3 | 0, -2 |

- The full payoff matrix is shown to Player 1, giving them a strategic advantage in planning their move.
- Player 2 must decide based on the observed action of Player 1, testing their ability to infer intent and optimize their response.

In our experiment, only Player 1's responses are collected for strategic reasoning evaluation.

Table 18: Incomplete Information Games Payoff Matrices

(a) Bayesian Coordination Games                    (d) Signaling Games

(b) Type 1                (c) Type 2                (e) Sender Payoffs                (f) Receiver Payoffs

| | | | | | | | | | | |
|---|---|---|---|---|---|---|---|---|---|---|
| 10, 10 | 5, 2 | | 8, 8 | 6, 3 | | 5, 5 | 2, 1 | | 4, 4 | 6, 3 |
| 7, 5 | 3, 3 | | 5, 4 | 2, 2 | | 3, 2 | 1, 0 | | 2, 3 | 1, 2 |

The incomplete information includes:

1. **Bayesian Coordination Games**: These games test an agent's ability to make optimal decisions under uncertainty. Each round presents two payoff matrices, one of which is randomly selected based on a given probability distribution. The player must make a decision without knowing which matrix is active, relying solely on probabilistic reasoning. The challenge lies in balancing risk and expected utility while considering the likelihood of each scenario. This game type evaluates how well an agent can infer optimal strategies from incomplete information.

2. **Signaling Games**: These games assess asymmetric information processing and strategic signaling. Player 1, the sender, has access to both a real and a fake payoff matrix. Player 2, the receiver, is only given the fake matrix and is aware that it does not reflect the true payoffs. Player 1 can send a signal to influence Player 2's decision, and Player 2 must determine whether to trust or disregard the signal. This game tests an agent's ability to strategically communicate and interpret signals in an environment where deception and trust play key roles.

Table 19: S-W 10

| | | |
|---|---|---|
| 47, 47 | 51, 44 | 28, 43 |
| 44, 51 | 11, 11 | 43, 91 |
| 43, 28 | 91, 43 | 11, 11 |

S-W 10 matrix is the matrix we borrow from the human research in behavioral economics in (54; 51).

# D Prompt Design

In this section, we present the complete prompts used in our experiments. Three types of experiments were conducted in all cases.

The first type involves the vanilla version of the prompts, with specific variations for the Sequential Game, Bayesian Coordination Game, and Signaling Game, as detailed in Appendix C. The second type includes zero-shot CoT prompts, which add two additional sentences to the base prompts.

For the demographic feature-embedded prompts, a template is followed, where demographic information is incorporated either as part of the system context or as an initial section of the user content, explicitly stating the user's profile with demographic features.

## D.1 Vanilla Experiment Prompt

The prompts used for the baseline evaluation of LLMs are listed in Table 20, 21, 22, and 23. These prompts are repeatedly sent using the LLMs' API or cloud services' API.

## D.2 Demographic Feature Embedding

The prompts used in the demographic-feature embedded experiment consist of two parts. The main body of the questions remains consistent with those in the vanilla prompt design. However, each prompt is augmented with a demographic component based on the template provided in Table 24. This demographic information is added at the beginning of each prompt to ensure that the LLM retains all features throughout the interaction, preventing memory loss during long-term conversations.

## D.3 Chain-of-Thought Embedding

The prompts used in the CoT embedding experiment are straightforward. Following the zero-shot CoT approach, a few additional sentences are appended to the end of each prompt to activate the CoT feature and enhance the reasoning capability of the model. Examples of the added CoT prompts are provided in Table 25.

Table 20: User Prompt for Competitive, Cooperation, and Mixed-motive Game

| User Prompt |
| --- |
| You are an assistant who helps to make a choice in a game. |
| You should make choices based on the idea of maximizing the number you get from a matrix. |
| Now you are player one/two. You have to pick a row/column number x/y from row 0, 1, and 2. |
| Then there will be a player two/one who will pick column number y/x from column 0, 1, and 2. |
| Your payoff will be the first/second value in location (x, y). |
| Assume the matrix is <A Matrix> |
| Please only give me a result of the row number you picked, do not include any thinking process. |

Table 21: User Prompt for Bayesian Coordination Game

| User Prompt |
| --- |
| You are an assistant who helps to make a choice in a game. |
| You should make choices based on the idea of maximizing the number you get from a matrix. |
| Now you are player one/two. You have to pick a row/column number x/y from row 0 and 1. |
| Then there will be a player two/one who will pick column/row number y/x from column 0 and 1. |
| Your payoff will be the first/second value in location (x, y). |
| With a <p> percent chance, you will be facing Matrix: <A Matrix>. |
| With a <1-p> percent chance, you will be facing Matrix: <B Matrix>. |
| Please only give me a result of the row number you picked, do not include any thinking process. |

Table 22: User Prompt for Sequential Game

| User Prompt |
| --- |
| Now you are player one. You are the first player to pick. You have to pick a row number x from row 0, 1, and 2. Then there will be a player two who will pick column number y from column 0, 1, and 2 based on your decision. Your payoff will be the first value in location (x, y). Assume the matrix is <A Matrix>. Please only give me a result of the row number you picked, do not include any thinking process. |

Table 23: User Prompt for Signaling Game

| User Prompt | |
| --- | --- |
| Player one | Player two |
| You should make choices based on the idea of maximizing the number you get from a matrix. Now you are player one. You have to pick a row number x from row 0 and 1. Then there will be a player two who will pick column number y from column 0 and 1. Your payoff will be the first value in location (x, y).

The true matrix that determines the payoff is Matrix: <Matrix True>.

However, the matrix player two will be seeing is Matrix: <Matrix Fake>.
Please only give me a result of the row number you picked, do not include any thinking process. | You should make choices based on the idea of maximizing the number you get from a matrix. Now you are player two. You have to pick a column number y from column 0 and 1. Then there will be a player one who will pick row number x from row 0 and 1. Your payoff will be the second value in location (x, y). The matrix you will be seeing is different from the true matrix, but you have to make your best selection based on your guess and the matrix you see.

The matrix is <Matrix Fake>.

Please only give me a result of the column number you picked, do not include any thinking process. |

Table 24: Demographic Feature Prompt Template

| Template |
| --- |
| Imagine a ['Age'] year old ['Gender'] with a ['Education'] degree, who is ['Marital Status'] and lives in a ['Location'] area. This individual identifies as ['Sexual Orientation'] and is ['Disability'], of ['Race'] descent, adheres to ['Religion'] beliefs, and supports ['Political Affiliation'] policies. Consider the risk preferences and decision-making processes of a person with these characteristics. |

Table 25: Chain-of-Thought version Prompt of Competitive Games

| User Prompt |
| --- |
| You are an assistant who helps to make a choice in a game. You should make choices based on the idea of maximizing the number you get from a matrix. Now you are player one/two. You have to pick a row/column number x/y from row 0, 1, and 2. Then there will be a player two/one who will pick column number y/x from column 0, 1, and 2. Your payoff will be the first/second value in location (x, y). Assume the matrix is <A Matrix> To decide, analyze the possible outcomes based on maximizing your payoff. Explain your reasoning step by step and then provide only the row/column number you picked as the final answer. |

# E Complete Reasoning Chains of Claude-3-opus and GPT-4o

**Claude-3-opus**

Let me understand the situation and analyze it step by step. First, I need to interpret the matrix you've provided. Let me organize it for clarity: Column 0 Column 1 Column 2 Row 0 (10,-10) (0,5) (-5,8) Row 1 (-10,10) (5,0) (8,-5) Row 2 (0,0) (5,-5) (-5,5) From your description, I understand that I am player one and need to pick a row number (x). Then player two will pick a column number (y). My payoff will be the first value in the location (x, y). Since my goal is to maximize my payoff, I need to consider how player two will respond to each of my choices. Let's analyze:

If I choose Row 0:

If player two chooses Column 0, my payoff is 10 If player two chooses Column 1, my payoff is 0 If player two chooses Column 2, my payoff is -5 Since player two wants to minimize my payoff, they would choose Column 2, giving me -5

If I choose Row 1:

If player two chooses Column 0, my payoff is -10 If player two chooses Column 1, my payoff is 5 If player two chooses Column 2, my payoff is 8 Player two would choose Column 0, giving me -10

If I choose Row 2:

If player two chooses Column 0, my payoff is 0 If player two chooses Column 1, my payoff is 5 If player two chooses Column 2, my payoff is -5 Player two would choose Column 2, giving me -5

Looking at the worst outcomes from each of my choices:

Row 0: -5 Row 1: -10 Row 2: -5

I should pick the row that gives me the maximum value among these worst-case scenarios. Both Row 0 and Row 2 give me -5 in the worst case, which is better than Row 1's worst case of -10. Between Row 0 and Row 2, I would choose Row 0 because if player two makes a mistake, I have the potential to get 10 (the highest possible payoff in the entire matrix), whereas with Row 2, the best I could get is 5.

Wait, I made a mistake in my analysis. Let me reconsider this problem using the concept of maximin strategy, which is appropriate for this game. For each row, I need to find the minimum payoff I could receive:

Row 0: min(10, 0, -5) = -5 Row 1: min(-10, 5, 8) = -10 Row 2: min(0, 5, -5) = -5

I should choose the row with the maximum of these minimum values. Both Row 0 and Row 2 have a minimum of -5, which is better than Row 1's minimum of -10. Since Row 0 and Row 2 are tied in terms of their minimum values, I'll look at their potential for higher payoffs. Row 0 has a potential maximum of 10, while Row 2 has a maximum of 5. Therefore, I would choose Row 0.

ORetryClaude does not have the ability to run the code it generates yet.Claude can make mistakes. Please double-check responses.

**GPT-4o**

We are Player Two (Column chooser) We pick a column y, then Player One picks a row x.

We get the second number from the chosen cell (x, y).

Now we assume Player One picks the row that minimizes our payoff, so again we apply maximin, looking at minimum second values per column:

Column 0: (0,0): -10 (1,0): 10 (2,0): 0 → Min = -10

Column 1: (0,1): 5 (1,1): 0 (2,1): -5 → Min = -5

Column 2: (0,2): 8 (1,2): -5 (2,2): 5 → Min = -5

So: Col 0: -10 Col 1: -5 Col 2: -5 → Maximin is -5, achieved by Col 1 or 2

Again, break tie with average:  Col 1 values:  5, 0, -5 → average = 0 Col 2 values: 8, -5, 5 → average = 2.67

So Column 2 is better.

