# OpenReview forum: "LLM Strategic Reasoning: Agentic Study through Behavioral Game Theory"
_NeurIPS.cc/2025/Conference — NeurIPS 2025 poster_

### Official Review · Reviewer_RAu2 · 2025-06-24

**Clarity:** 4
**Significance:** 3
**Originality:** 3
**Rating:** 4
**Confidence:** 3

**Summary:**

This paper introduces a behavioral game-theoretic framework to evaluate the strategic reasoning capabilities of large language models (LLMs). Moving beyond traditional Nash Equilibrium (NE) benchmarks, the authors apply a Truncated Quantal Response Equilibrium (TQRE) model to estimate the reasoning depth (τ) of 22 state-of-the-art LLMs across 13 one-shot matrix games, including competitive, cooperative, mixed-motive, and belief-based games. Results show that reasoning depth varies not only across models and game types but also depends on reasoning styles and context. Chain-of-thought (CoT) analysis reveals that longer reasoning does not always imply better decisions. Furthermore, embedding demographic personas affects performance, with some models exhibiting bias when assigned minority identities. Overall, the study highlights that strategic decision-making in LLMs is shaped by both architecture and prompt context, and that performance alone is not sufficient to ensure fairness or human-aligned behavior.

**Questions:**

A question that’s closely related—but perhaps beyond the current scope of the paper—is whether the **τ parameter systematically varies with the complexity of the task structure**. From the provided tables, it appears that τ does differ across tasks. However, a deeper and more important question is: **does each model exhibit consistent behavioral adaptation across task types**? That is, does a model tend to use simpler inference in simpler games, and engage in deeper reasoning when the task demands it? And is this reflected in its τ estimates across contexts?

Answering this would be critical for understanding whether a model’s strategic reasoning is **generalizable and context-sensitive**, or merely a reflection of **fixed priors shaped by uneven training exposure**. If the model truly possesses adaptable reasoning capabilities, we should see its behavior align with the structural demands of different tasks. Otherwise, its performance may just reflect biases in pretraining, rather than a principled ability to engage in strategic thinking.

**Ethical Concerns:**

["NO or VERY MINOR ethics concerns only"]

**Final Justification:**

The authors partially addressed my concerns about empirical evidence and I would like to increase my rating to 4 to support the quality and originality of the paper.

**Limitations:**

Yes.

**Paper Formatting Concerns:**

No.

**Quality:**

3

**Strengths And Weaknesses:**

### **Strengths**
- This paper introduces a diverse and representative set of LLMs and tasks from game theory (e.g., complete vs. incomplete information, cooperative vs. competitive vs. sequential). The evaluation setup is comprehensive and covers a wide range of strategic scenarios.
- The theoretical foundation and modeling framework are solid. The use of TQRE to estimate each model’s reasoning depth (τ) from one-shot behavior is well-motivated and connects nicely to behavioral game theory.
- I also appreciate the ethical and demographic analyses. The study probes how demographic prompts (e.g., gender, sexuality) affect strategic behavior, which raises important implications for fairness, bias, and safe AI deployment in interactive settings.

### **Weaknesses**
- I’m a bit skeptical about interpreting the τ parameter strictly as “reasoning depth.” While the framework is theoretically sound, the evaluation is entirely based on one-shot decisions. That means the models are relying on prior knowledge or inductive biases, not adapting to opponents. So instead of saying one model reasons more deeply than another, I’d frame it as *being biased toward deeper recursion by default*. Without interaction or adaptation, it’s hard to claim that actual recursive reasoning is happening.
- Another limitation is that the paper focuses mostly on behavioral outcomes. The chain-of-thought (CoT) content is only lightly touched upon. Since recursive reasoning could be more directly observable in the CoT traces, a more systematic analysis there would strengthen the claims. For example, parsing the CoTs for syntactic recursion, or using an LLM-as-judge to label reasoning levels, could reveal whether verbal reasoning depth aligns with behavioral τ. This would help ground the τ measure and improve interpretability.
- Another missing point is the authors did not report how well the models (TQRE) fit to each LLM's behaviors (as well as in a diverse of tasks). Without knowing how well the TQRE model explains the behaviors of LLMs, it's hard to tell convinceness of the tau estimation. Another one is to figure out whether TQRE is always the best model to explain the LLM's behaviors. The authors should prepare some classical behavioral game theory or theory of mind models for testing against TQRE. Those model comparison and goodness-of-fit metrics would help to improve the convinceness of the parameter results.

---

> ### Author Rebuttal · Authors · 2025-07-31
>
> Thank you for the review. Below are our responses to your questions and comments.
>
>
>
>
> **Weakness 1 Response:**
>
> The concept of reasoning depth (sometimes also referred to as “levels of reasoning” or “strategic sophistication”) is a longstanding and standard practice in behavioral game theory and experimental economics. Classic works (e.g., Camerer, Ho, & Chong 2004; Costa-Gomes, Crawford, & Broseta 2001; Wright & Leyton-Brown 2010) use one-shot games specifically to benchmark and quantify these capacities.
>
> Secondly, dynamic, multi-stage interactions are commonly analyzed as sequences of one-shot decisions, each defined by the agent’s reasoning given the available information at that stage. This decomposition is a standard approach in game theory and behavioral economics (Camerer 2003; Fudenberg & Tirole 1991), where each round is modeled as a one-shot decision conditioned on the current information set. Similar to current practice with LLMs, multi‑round analyses are implemented by appending the transcript of each preceding round to the prompt, allowing the model to revise its decisions and adapt to its opponent. From the LLM’s perspective, a dynamic strategic game thus remains a single‑shot decision problem presented within an ever‑growing context window. Although the model’s finite context length eventually limits how many rounds can be accommodated, this constraint does not undermine our framework: the core evaluation metric, reasoning depth, captures the model’s capabilities across all relevant dimensions and therefore remains valid regardless of prompt length.
>
> Adaptation and learning over repeated play are valuable research avenues, but they differ from the baseline strategic reasoning that our framework measures. Recursive reasoning asks, “How many steps ahead can the agent think, given its priors?” Learning dynamics ask, “How efficiently and quickly can it update those priors?” Our framework targets the former. Moreover, whether the agent relies on innate priors or on newly acquired knowledge of the game, its “thinking paradigm” remains the same.
>
>
>
> **Weakness 2 Response:**
>
> Thanks for the suggestions. Adding comparison using syntactic recursion for CoT parsing or using LLM-as-judge are promising ideas, and we will definitely consider them in our ongoing and future studies. Nonetheless, we want to point out that we opted for experienced human annotators in the current study for greater interpretability and established validity, and because there is currently no robust evidence that LLMs can accurately or validly assess the reasoning depth of other LLMs. Employing LLMs as judges would require substantial new validation and would shift the focus of our work away from behavioral evaluation and interpretability. In current work, using LLMs to judge other LLMs is challenging also because it could introduce circular logic, as both may share training data, inductive biases, or flaws. We agree that using additional analysis is value-added, but it can only be considered as parallel analysis.
>
> CoT is a major focus of our work. In this work, we systematically collected diverse reasoning traces from each model and anonymized them for “blind review.” Our researchers independently classified the reasoning styles using a structured coding rubric, with cross-validation to ensure reliability. The procedure directly links behavioral patterns (e.g., “self-interested” or “distrustful” models struggling to cooperate) to CoT explanations. Please check Section 4.2 for more details.
>
> We are happy to include your proposed judges for comparison.
>
>
> **Weakness 3 Response:**
>
> There might be a misunderstanding here. The TQRE model used in our analysis is not a data-driven machine learning model, but a well-established structural model from behavioral economics and experimental game theory. TQRE does not “learn” its parameters from the data in a flexible way, but rather imposes a theoretically motivated structure to interpret observed choices, which is an approach widely used for both human and agent behavior. If you are referring to the maximum log likelihood of TQRE fits, for individual models and games, the average negative maximum log likelihood per game is typically in the range of 0–2, which is comparable to prior experimental studies (Camerer, Ho, & Chong, 2004).
>
> We adapted TQRE as the foundation of our quantitative framework based on rigorous justification. It builds upon and extends earlier behavioral game theory models, such as the Cognitive Hierarchy model families and the Quantal Response Equilibrium models, by incorporating bounded rationality, noisy response, and empirically realistic distributions of reasoning depth. It is already the most comprehensive and realistic model compared to other CH-family and QRE models, as shown in comparative studies (e.g., Wright & Leyton-Brown, 2010, AAAI), to provide the best fit to actual agent behavior.
>
>
>
> **Questions Response:**
>
> We address precisely this issue in Section 4.2 of our paper, where we analyze model behavior and internal reasoning processes across a diverse set of game types. Our results show that models do exhibit typical patterns in different types of games, and each model tends to have its own areas of strength and weakness given their reasoning styles, and these patterns are reflected in both their behavioral choices and internal chain-of-thought reasoning.
>
> It is important to recognize that the “task complexity” you refer to is not an absolute measure. For instance, a cooperative game might be straightforward for a model with a preference toward trust, but much more complex for a model that assumes opponents are untrustworthy, requiring more layers of strategic reasoning and anticipation of adversarial play.
>
> The TQRE framework is specifically designed to accommodate this heterogeneity. Rather than assuming a fixed hierarchy of task difficulty, TQRE estimates τ based on how an agent actually navigates the strategic landscape of each game, given its own reasoning process, beliefs, and biases. This means that an increase in τ for a particular agent in a given game reflects that agent’s need for deeper recursion, and not an externally imposed notion of complexity.
>
> Our findings confirm that models display distinct patterns across game types rather than a universal progression from “simple” to “complex.” This agent-relative perspective is fundamental in behavioral game theory and essential for interpreting τ as a measure of context-sensitive strategic depth.
>
>
>
>
> **Reference:**
>
> Wright, J., & Leyton-Brown, K. (2010, July). Beyond equilibrium: Predicting human behavior in normal-form games. In Proceedings of the AAAI Conference on Artificial Intelligence (Vol. 24, No. 1, pp. 901–907).
>
> Camerer, C. F., Ho, T.-H., & Chong, J.-K. (2004). A cognitive hierarchy model of games. Quarterly Journal of Economics, 119(3), 861–898.
>
> Costa-Gomes, M. A., Crawford, V. P., & Broseta, B. (2001). Cognition and behavior in normal-form games: An experimental study. Econometrica, 69(5), 1193–1235.

---

> > ### Comment · Reviewer_RAu2 · 2025-08-01
> >
> > **Thanks to the authors for the thoughtful and effortful rebuttal.**
> >
> > Regarding **Weakness 1**, I agree with many of the points raised—particularly that one-shot settings have their own value and contribute meaningfully to understanding agent reasoning. However, I believe the current experimental design captures only a partial view of LLMs' strategic learning capabilities. Given their pretrained priors, LLMs can exhibit reasoning about how agents would react. In repeated games, we are not limited to examining only the first trial, as done in the current study; rather, we can also investigate dynamic interactions and how reasoning evolves through interaction. These two perspectives are not mutually exclusive—rather, the latter can serve as a valuable supplement to the former. While I acknowledge the limitations of in-context learning as context length increases, adding 10 or even 50 rounds of data remains well within the capacity of most models tested in this study.
> >
> > Regarding **goodness-of-fit**, it would be helpful if the authors could provide a summary table listing, for tasks with varying numbers of options, the average likelihood and SEM (or 95% confidence intervals) for each model. Additionally, reporting accuracy—e.g., via greedy decoding of the likelihood—would provide further insights into model performance.
> >
> > Regarding **model comparison**, I recognize that TQRE has theoretical foundations in prior literature. However, to my knowledge, this theory was not originally developed based on LLM behavioral data (e.g., from the 2010 paper referenced). It would be more convincing to supplement the theoretical rationale with empirical evidence from the current dataset. Specifically, fitting a broader range of model candidates could help assess whether TQRE is indeed the most appropriate framework for interpreting LLM strategic behavior.
> >
> > I generally agree with the other points raised in the rebuttal. However, the three points above—particularly the two empirical concerns—remain central limitations of the current evaluation. Therefore, I am unable to raise my score at this time without further detailed empirical evidence.

---

> > > ### Author Response · Authors · 2025-08-02
> > >
> > > Thank you for your quick response to our rebuttal. We are happy to offer further clarification to address potential misunderstandings and add additional empirical evidence.
> > >
> > > 1. Regarding Weakness 1:
> > >
> > > We agree that dynamic settings are an important next step. We believe that in the future, it will be a much bigger picture in LLM strategic evaluation. As this is still an emerging area, our goal with this paper is to provide a clear, foundational story: using a human-originated behavioral model to benchmark LLM capabilities for the first time systematically. One-shot strategic decision-making forms the core of LLM evaluation, so we began with multiple models and a wide range of games to establish a baseline of their abilities.
> > >
> > > Introducing more complex, multi-round tasks at this early stage could risk obscuring key patterns rather than clarifying them, especially given that many models still struggle with the basics. The fact is well-reasoning models are a minority, making games that are even more complex will not benefit the evaluation. To make a useful comparison here and as an initial touch of multi-rounds, one of our game settings is a sequential game, which involves two rounds, using the first mover’s action to set the stage for the second.
> > >
> > > We also look forward to the development of this area and seeing complex and multiple rounds of agent interactions.
> > >
> > > 2. Regarding goodness-of-fit:
> > >
> > > As we provided in previous rebuttal contents, “maximum log likelihood of TQRE fits during the MLE process, for individual models and games, the average negative maximum log likelihood per game is typically in the range of 0–2, which is comparable to prior experimental studies”
> > >
> > > We are more than happy to add this table (attached as a preview) to the revised version, and we believe that this will add more transparency. One thing we still want to clarify is that TQRE is a structural model rather than a flexible machine learning model; standard fit metrics are interpreted in terms of their theoretical rather than predictive properties.
> > >
> > > | Scenario | claude-3 | gemma | internlm 2.5 | LLAMA 405B | Llama 8B  | gpt o1 | gpt o3 mini | Qwen 32 B  |
> > > | :-------------------------- | ---------: | ---------: | -----------: | ---------: | --------: | --------: | ----------: | ---------: |
> > > | **Competitive**  | -2.19722465 | -2.04506862 | -2.16384241 | -1.66953973 | -2.19700513 | -1.84922998 | -1.42741863 | -2.16181531 |
> > > | **Cooperation** | -1.17692360 | -1.38629437 | -1.38629438 | -1.38629438 | -1.38629441 | -1.30439678 | -1.25332626 | -1.35923763 |
> > > | **Incomplete-Bayesian-p05** | -0.49005370 | -1.37538293 | -1.38629439 | -1.38149054 | -1.36696411 | -0.02325751 | -0.02325751 | -1.27305147 |
> > > | **Incomplete-Bayesian-p09** | -0.02325801 | -1.37929141 | -1.33920182 | -1.37257037 | -1.38184227 | -0.02325801 | -0.02325801 | -0.98713199 |
> > > | **Incomplete-Signaling**    | -0.93369680 | -1.32586959 | -1.38612032 | -1.38629582 | -1.37328338 | -0.80059012 | -1.33583277 | -0.63836366 |
> > > | **Mix motive** | -1.38629472 | -1.38629440 | -1.38629516 | -1.38629483 | -1.38629507 | -1.36847250 | -1.25581520 | -1.28123369 |
> > > | **Sequential** | -0.65744618 | -0.67508254 | -0.90262492 | -0.69275761 | -0.99635401 | -0.74234520 | -0.77996237 | -1.08046524 |
> > > | | |  |  |  | |  |  | |
> > >
> > > 3. Regarding model comparison:
> > >
> > > You are correct that the TQRE model was not originally developed from LLM behavioral data, as LLMs have only emerged in recent years.
> > >
> > > However, it is important to clarify that the models discussed, Nash Equilibrium, QRE, CH, and TQRE, are not independent or parallel alternatives. Rather, there is a clear historical and theoretical progression:
> > >
> > > As detailed in Sections 2, 3, and the appendix: Nash Equilibrium is the theoretical origin, based on perfect rationality. QRE and CH extend Nash by relaxing its most restrictive assumptions, each supported by substantial empirical evidence demonstrating improved real-world fit. Over time, other behavioral models have emerged, but many were not widely adopted due to theoretical or empirical limitations. TQRE combines the advantages of both QRE (allowing quantal response) and CH (allowing heterogeneity in reasoning levels), resulting in a unified framework that subsumes both models.
> > >
> > > To emphasize: the models mentioned above, which have appeared in prior LLM work, are in fact special cases of TQRE. For instance, as parameters approach certain limits, TQRE converges to QRE or CH. In our empirical evaluation, we did not observe these limiting cases; the data fit best within the broader TQRE parameterization. There is a hierarchical structure: TQRE encompasses prior models within its framework, analogous to how a general mathematical model includes earlier, more restrictive versions as special cases.
> > >
> > > **Regardless of the score, we hope these clarifications help resolve any misunderstandings and provide a clearer view of our contributions. Thank you again for your time and consideration.**

---

> > > > ### Comment · Reviewer_RAu2 · 2025-08-07
> > > >
> > > > Dear Authors,
> > > >
> > > > Sorry for my late reply since I have been traveling. I appreciate your efforts in rebuttal and your provided new results. I partially agree that TQRE is a well-constructed theoretical model, and other common models are kinds of simplified versions of TQRE. However, this does not contradict inquiring about model comparisons in your LLM dataset. I would believe this could be a strong plus to the current results and won't take down the contribution of the TQRE model itself.
> > > >
> > > > Regarding the supplemented results, I would, in principle, increase my rating to 4 given a more comprehensive result. However, to avoid any confusion, could the authors please explain in each scenario how many options there are (or whether they might be somehow mixed)? This could help to establish the baseline of chance-level. As far as I see, for cooperation scenarios, Table 1 suggests it is a two-option problem, and thus the chance level log-likelihood should be around 0.69. Your current results suggest the fitting is indeed worse than chance level, significantly. So I would suggest more clarification on the results, so we can correctly evaluate these results. Thanks!

---

> ### Author Response · Authors · 2025-08-07
> **Further clarifications**
>
> Dear Reviewer,
>
> Thanks for your engagement during travel, and for your careful follow-up. We are happy to provide further clarification on the confusion:
>
> Chance-level baseline: In each scenario, we compute log-likelihoods over the joint action profile for simultaneous games, because in each trial we recorded both the row‐player’s and column‐player’s binary choice. For example, for cooperation game, this means the chance-level log-likelihood is ln  0.25 = –1.386, not –0.693 (which would be for a single binary choice). We will clearly state this baseline in the revised tables.
> For further interpretation, values near the baseline are consistent with spontaneous or low-level play (level-0 reasoners). We agree that interpreting in conjunction with model parameters can give a fuller picture of agent behavior.  We will include a detailed appendix table showing these action counts, chance baselines, and model fits for each scenario, along with explanatory note. We appreciate your feedback and agree this will make the results more transparent and easier to interpret.
>
> We also appreciate the suggestion on complementary model comparisons. We will add this to our project pipeline and look forward to exploring these extensions.
>
> We hope these explanation resolves any confusion, and thank you again for your feedbacks, we truly appreciate your engagement and suggestions!

---

> > ### Comment · Reviewer_RAu2 · 2025-08-07
> >
> > Thanks to the authors for quick feedback! I would like to increase the score to 4 in response to the supplemented results and clarification.

---

### Official Review · Reviewer_aEWT · 2025-06-30

**Clarity:** 3
**Significance:** 3
**Originality:** 3
**Rating:** 4
**Confidence:** 3

**Summary:**

This paper presents a framework for evaluating the strategic reasoning capabilities of Large Language Models (LLMs) when they act as agents in interactive scenarios. The authors argue that traditional evaluation methods, often based on Nash Equilibrium (NE), are insufficient as they overlook the bounded rationality and stochasticity inherent in LLM decision-making. To address this gap, the paper introduces an evaluation framework grounded in behavioral game theory.

The core of the methodology is the application of the Truncated Quantal Response Equilibrium (TQRE) model to analyze LLM behavior across a diverse library of 13 strategic games. By fitting the TQRE model to the observed choices of 22 state-of-the-art LLMs, the framework estimates a key parameter, τ, which quantifies the depth of strategic reasoning for each model. The study further investigates how reasoning capabilities are influenced by Chain-of-Thought (CoT) prompting and the assignment of demographic personas. The key findings are threefold: (1) Reasoning depth varies significantly across models and is not solely determined by model size; (2) Different models exhibit distinct reasoning styles (e.g., maximin, belief-based); and (3) LLMs display context-sensitive shifts in reasoning and potential social biases when assigned demographic identities.

**Questions:**

I would be willing to raise my score if the authors could elaborate on the following points.

- The analysis of dominant reasoning styles is fascinating. Are these styles stable, intrinsic properties of the LLMs, or are they highly context-dependent? For example, would a model like GPT-01, which shows a strong maximin preference, adapt its style in a game with more cooperative elements or under prompts that encourage trust? A deeper exploration of the malleability of these styles would be very insightful.
- The discovery of reasoning shifts based on demographic personas is a critical finding. The paper shows that it happens, but the why remains an open question. Do the authors have a hypothesis regarding the mechanism? Is it primarily a reflection of statistical biases in the training data (e.g., text corpora associating certain demographics with certain behaviors), or could the RLHF process also play a role in shaping these context-dependent responses?
- The framework estimates both τ (depth) and γ (precision/noise). The main analysis focuses on τ. Is there any interesting correlation or trade-off between τ and γ across models? For instance, do models with deeper reasoning also tend to be less noisy (higher γ)? Did the persona-based prompts also cause significant shifts in γ, which could distinguish between a change in "cognitive ability" versus a change in "decision consistency"?
- The qualitative analysis relies on the reasoning chains (CoT) generated by the models. How confident can we be that these chains accurately reflect the model's "true" internal reasoning process, rather than being a post-hoc rationalization of a decision reached through other means? While this is a notoriously difficult problem, a brief discussion of this caveat would add to the paper's rigor.

**Ethical Concerns:**

["NO or VERY MINOR ethics concerns only"]

**Final Justification:**

After engaging in a thorough and productive discussion with the authors, I have decided to raise my score to "Borderline Accept". My decision is primarily driven by the paper's originality and its potential to establish a new paradigm for evaluating strategic reasoning in Large Language Models.

The primary strength of this work is its pioneering contribution. It moves beyond traditional, often rigid, evaluation metrics by introducing a sophisticated framework grounded in behavioral game theory. The TQRE model and the resulting τ (reasoning depth) metric offer a nuanced, interpretable, and quantitative way to assess LLM behavior.

Through our discussions, several of my initial concerns were effectively addressed. The authors clarified that the Chain-of-Thought (CoT) analyses are based on concurrent reasoning traces, not post-hoc rationalizations, which strengthens the validity of their qualitative findings. They also provided a well-reasoned argument for the theoretical standing of the TQRE model within the broader family of behavioral models, justifying its selection as a foundational element of their framework.

While the current framework is robust, I believe future iterations could be further enriched. For instance, a deeper dive into the γ (precision/noise) parameter could uncover additional layers of insight into decision consistency, providing a valuable complement to the current focus on reasoning depth. Likewise, while the theoretical justification for TQRE is strong, supplementing it with direct empirical model comparisons on LLM-generated data would offer the most conclusive validation for its application in this novel domain.

In my final assessment, I assign the greatest weight to the foundational value and forward-looking impact of this research. The paper provides a framework to understand the 'what' and 'why' of LLM strategic behavior, which is a contribution before the community can effectively address the 'how' of improving it. For this reason, I believe the compelling reasons to accept this paper, centered on its novelty and foundational significance, now outweigh these remaining areas for future methodological refinement.

**Limitations:**

The authors have included a "Limitations and Future Directions" section, which is commendable. To re-emphasize and structure these points:
- L1: Focus on One-Shot, Static Games: As mentioned, the current framework is designed for and evaluated on static, normal-form games. Its direct application to dynamic, extensive-form games where the game state evolves and players' histories matter is a non-trivial extension and a key limitation of the current work's scope.
- L2: Causality of Prompting Effects: The paper demonstrates strong correlations between prompts (CoT, personas) and changes in reasoning behavior. However, as the authors note, establishing a clear causal link remains an open challenge. The "why" behind these effects is not fully explained by the current framework.
- L3: Semi-Dynamic Framework: The current evaluation is "semi-dynamic" as it tests pre-trained models. Extending the framework to analyze real-time multi-agent interactions, where LLMs learn and adapt to each other over multiple rounds, is a significant future step and a limitation of the current static analysis.

**Paper Formatting Concerns:**

The paper adheres to all the formatting requirements specified in the NeurIPS 2025 Paper Formatting Instructions.

**Quality:**

3

**Strengths And Weaknesses:**

**Strengths:**
- The introduction of a new paradigm for evaluating LLM strategic reasoning. Moving beyond the binary pass/fail assessment of Nash Equilibrium, the framework provides a continuous, quantitative metric (τ) for reasoning depth, which is grounded in cognitive science from behavioral game theory. This allows for a more interpretable, and fine-grained analysis of LLM capabilities.
- The paper is  well-written, making complex concepts from behavioral game theory accessible to a broader machine learning audience. The research provides a potentially useful tool for researchers and practitioners to audit and understand LLM behavior in interactive settings.

**Weaknesses:**

- Reliance on a Specific Behavioral Model (TQRE): The entire framework's quantitative conclusions are predicated on the Truncated Quantal Response Equilibrium (TQRE) model. While TQRE is a well-established model in behavioral economics, the paper's findings are inherently conditional on its assumptions (e.g., Poisson-distributed reasoning levels). The work could be strengthened by acknowledging this model-dependency more explicitly and perhaps discussing how results might change under alternative behavioral models (e.g., Cognitive Hierarchy with different belief formation rules).
- Limited Scope of Interaction: The study focuses exclusively on one-shot, normal-form (matrix) games. This is a deliberate and effective choice for isolating initial strategic reasoning. However, it does not capture the complexities of dynamic, multi-stage interactions where learning, adaptation, reputation, and state evolution play critical roles. This limits the direct generalizability of the specific τ values to more complex, extensive-form games.

---

> ### Author Rebuttal · Authors · 2025-07-31
>
> Thank you for the review. Below are our responses to your questions and comments.
>
>
>
>
> **Question 1 Response:**
>
> Our experiments suggest that dominant reasoning styles are often intrinsic and stable properties of the LLMs we tested. As you mentioned in the question, GPT-o1’s strong maximin preference remains the dominant pattern across most of the 13 games in our library, regardless of specific game context. This intrinsic tendency also explains why GPT-o1 tends to underperform in cooperative or mixed-motive games; it still chooses the secure but collectively inferior outcome.
>
> Our goal is to benchmark the models’ native strategic reasoning. Supplying detailed guidance would blur that line, turning the task into “follow‑the‑instructions” rather than revealing the model’s default reasoning style. We therefore keep prompts minimal and identical across models.
>
>
> **Question 2 Response:**
>
> Thanks for bringing up this question. We acknowledge this as an important and interesting problem. The proprietary training corpora and RLHF pipelines of most models are opaque, so we cannot trace a single, provable cause directly.  Any updates to the weights of the model can and will introduce bias to the model’s behavior, and it is quite difficult to pinpoint where the bias is introduced. What we can say, based on our additional “internal‑thinking” audit, is the following:
>
>
>     If we ask the model directly, “Will you treat personas differently?” it would most likely deny the answer. However, the observation would still show the shifts in responses. This suggests that this bias is internal and implicit.
>
>     The most plausible hypothesis is that large-scale text corpora encode stereotypical associations between demographic and behavioral cues. These priors affect downstream decisions even though the explicit request to the model or its rationale is “to be fair”. For example, some cultures have an emphasis on unity and moderation, so the model trained heavily using data in those regions might benefit more from a cooperation game, such as Deepseek. RLHF can either dampen or amplify these latent patterns by bringing researchers’ personal “preferences”. Without access to the non-public data, it’s hard to disentangle these factors.
>
>
> The aim of this work is therefore not to locate the ultimate cause of the bias, but to demonstrate that the framework is sensitive and robust enough to reveal such latent biases, even when these trends are absent from the models’ verbal justification.
>
> **Question 3 Response:**
>
> In our analysis, we primarily focus on τ as it directly captures strategic reasoning depth, which is central to our study. γ, on the other hand, is included mainly as a control variable to account for variability and contextual confounders in decision-making. We will add further clarification to the final draft.
>
> Regarding the correlation or trade-off between τ and γ, we did not observe any strong or consistent patterns across models or games in our results, and our data does not support a simple relationship or trade-off between these parameters. Since γ did not reveal consistent or generalizable patterns across models, we chose to focus our main discussion on τ in order to maintain clarity and prioritize our primary research objectives within the available space. We thank the reviewer for highlighting this interesting question, and we will consider a deeper analysis of more complex relationships between these two parameters in future work.
>
> **Question 4 Response:**
>
> The reasoning chains were obtained directly from the model outputs using the API feature that returns the model’s internal step-by-step reasoning for each decision, rather than asking for an explanation after the fact. This process is analogous to revealing the model’s “thought process” as it happens, so it’s not a post-hoc response. Thank you for raising that up, and we will add a note to the paper to make this point clearer.
>
> **Weakness 1 Response:**
>
> Our quantitative framework is grounded in TQRE because, in comparative studies such as Wright & Leyton-Brown (2010, AAAI), it has been shown to fit human data better than alternative models—including Cognitive Hierarchy and standard Quantal Response Equilibrium.
>
> TQRE is closely related to the CH family of models, which can include a variety of belief formation rules. Our use of the Poisson distribution in TQRE follows both theoretical precedent and empirical findings that most agents reason at lower levels, with fewer at higher depths (Camerer, Ho, & Chong, 2004; Costa-Gomes, Crawford, & Broseta, 2001; Stahl & Wilson, 1994, 1995).
>
> We will discuss how future work could incorporate and compare alternative models into the final draft. Thank you for the suggestion.
>
> **Weakness 2 Response:**
>
> Our methodological choice is grounded in the perspective that dynamic, multi-stage interactions can be decomposed into rounds of one-shot decisions, with each round representing a cross-sectional snapshot of agent reasoning given the available information at that stage.
>
> For example, consider a multi-round coordination game:
>
> In the first round, agents have no history or prior information about each other, so they make decisions based solely on the game’s structure and any available cues, just the same as in the one-shot coordination game. In the second round and beyond, agents update their beliefs based on observed past actions. These historical interactions then become part of the information available for subsequent decisions. Thus, each new round can still mirror as a one-shot decision, but conditioned on a richer informational context.
>
>
>
> **Reference:**
>
> Wright, J., & Leyton-Brown, K. (2010, July). Beyond equilibrium: Predicting human behavior in normal-form games. In Proceedings of the AAAI Conference on Artificial Intelligence (Vol. 24, No. 1, pp. 901–907).
>
> Camerer, C. F., Ho, T.-H., & Chong, J.-K. (2004). A cognitive hierarchy model of games. Quarterly Journal of Economics, 119(3), 861–898.
>
> Costa-Gomes, M. A., Crawford, V. P., & Broseta, B. (2001). Cognition and behavior in normal-form games: An experimental study. Econometrica, 69(5), 1193–1235.
>
> Stahl, D. O., & Wilson, P. W. (1994). Experimental evidence on players' models of other players. Journal of Economic Behavior & Organization, 25(3), 309–327.
>
> Stahl, D. O., & Wilson, P. W. (1995). On players' models of other players: Theory and experimental evidence. Games and Economic Behavior, 10(1), 218–254.

---

> > ### Comment · Reviewer_aEWT · 2025-08-03
> > **Official Comment by Reviewer aEWT**
> >
> > I would like to thank the authors for their detailed rebuttal. The clarifications on several key aspects of the study, such as the stability of the models' intrinsic reasoning styles, the plausible sources of persona-based biases, and the concurrent nature of the CoT generation, have certainly strengthened the manuscript.
> >
> > However, despite these helpful explanations, two concerns about the core of the evaluation framework persist.
> >
> > First, my primary reservation lies with the incomplete treatment of the TQRE model's parameters. The rebuttal dismisses the γ (precision/noise) parameter as a simple control variable, which I believe understates its critical behavioral significance. This parameter is essential for understanding an agent's sensitivity to incentives and the consistency of its decision-making. By omitting a detailed analysis of γ, the paper misses a crucial opportunity to disentangle whether the observed behavioral shifts stem from a change in cognitive depth (τ) or from a change in decision certainty (γ). This distinction is not trivial; it is central to a complete understanding of LLM strategic behavior. A framework that claims to provide fine-grained analysis should not set aside one of its core explanatory variables.
> >
> > Second, there is an unresolved question about the suitability of the TQRE model itself for evaluating LLMs. The TQRE framework, and the broader family of Level-k models, were developed to model human bounded rationality, which is often characterized by iterative, step-by-step reasoning. The paper's reliance on a model designed for human cognition without empirically validating its appropriateness for LLMs (for example, by comparing its fit against other behavioral models on this new dataset) leaves the foundational assumptions of the work underexplored.
> >
> > I look forward to further discussion with the authors on these matters.

---

> ### Author Response · Authors · 2025-08-04
>
> We are pleased to have addressed your earlier questions. As you mentioned a willingness to raise your score if concerns were addressed, we hope our responses provide the needed clarity:
>
> **1. Regarding role of gamma**
>
> Thank you for your concern. Our analytical approach is well-grounded in established behavioral modeling practice, where γ is routinely treated as a control parameter. This is because γ does not capture a singular, interpretable behavioral trait; rather, it absorbs a wide array of influences, including individual quirks, random variability, and context-driven noise. It tries to encapsulate and isolate the environmental confounding factors that may affect evaluating the τ (reasoning depth). Its main function is to ensure that estimates of τ are not conflated with such noise, thereby allowing τ to provide a clear and interpretable signal of cognitive sophistication.
>
> Given our primary research objective to quantify and compare the reasoning depth of LLMs, it is methodologically appropriate to focus on τ as the most direct and meaningful index of strategic sophistication. Including γ as a control allows us to capture robust and unbiased estimation of τ, while avoiding misinterpretation of behavioral inconsistency as shallow reasoning.
>
> For completeness and transparency, we provide a summary table of γ across models and tasks here, which will also be added to the appendix, which further clarifies its role in the revised manuscript. We could have spent more paragraphs discussing the impacts of γ with these data. Still, we believe the current approach allows our work to provide a focused and rigorous contribution, telling one clear story about reasoning depth, without introducing other ambiguity.
>
> | Scenario | claude-3 | deepseek 2.5 | deepseek r1 | gemini 2.0 | gemma | gpt 4o | gpt o1 | gpt o3 mini | internlm 2.5 | LLAMA 405B | Llama 8B | Qwen 32 B |
> | :--- | ---: | ---: | ---: | ---: | ---: | ---: | ---: | ---: | ---: | ---: | ---: | ---: |
> | competitive | 5.833 | 7.669 | 11.383 | 1.719 | 8.834 | 2.053 | 0.441 | 1.131 | 5.384 | 6.154 | 0.000 | 0.370 |
> | cooperation | 1.687 | 3.347 | 0.348 | 2.009 | 0.348 | 1.667 | 3.349 | 3.347 | 1.667 | 1.667 | 0.336 | 1.696 |
> | Incomplete-Bayesian-p05 | 5.000 | 5.001 | 0.000 | 5.001 | 0.031 | 0.846 | 5.001 | 5.001 | 0.000 | 0.025 | 0.337 | 6.609 |
> | Incomplete-Bayesian-p09 | 5.276 | 5.276 | 0.014 | 5.276 | 0.317 | 0.950 | 5.276 | 5.276 | 0.593 | 0.441 | 5.861 | 0.842 |
> | Incomplete-Signaling | 0.355 | 0.120 | 0.156 | 0.300 | 0.129 | 0.236 | 0.395 | 0.120 | 0.061 | 0.000 | 0.081 | 0.882 |
> | mix motive | 0.000 | 1.711 | 0.000 | 0.348 | 0.000 | 0.000 | 3.717 | 1.720 | 0.000 | 0.000 | 0.000 | 1.813 |
> | Sequential | 38.675 | 52.656 | 40.615 | 0.000 | 52.362 | 34.198 | 0.676 | 35.734 | 42.306 | 37.102 | 48.384 | 48.972 |
> |   |   |   |   |   |   |   |   |   |   |   |   |   |
>
> **2. Thank you for raising this question. We hope to fully clarify our modeling rationale:**
>
> First, our qualitative analysis in Section 4.2 demonstrates that LLMs frequently display step-by-step, multi-level reasoning in their chain-of-thought outputs. This evidence directly supports the relevance and interpretability of behavioral models like TQRE, for characterizing LLM decision-making, not just human reasoning.
>
> Second, our choice of TQRE is under rigorous justification. In behavioral game theory models, Nash Equilibrium, QRE, CH, and TQRE form a clear theoretical lineage, not a set of parallel alternatives. TQRE combines the advances of QRE and CH, and has consistently shown superior empirical fit across human data and LLM data. Previous works have demonstrated the justification of applying and evaluating earlier models like NE to LLMs, as we have pointed out that these earlier models are all special or limiting cases within the broader TQRE framework, making the TQRE framework a comprehensive and robust choice for unified evaluation.
>
> Third, we do not assume LLMs are identical to humans in cognition. Instead, we use TQRE as a principled, interpretable benchmark, enabling meaningful comparison and insight into agentic reasoning across domains. Our empirical findings further support this choice: stepwise reasoning is observed, and the model structure captures the main behavioral regularities in LLM play.
>
> We believe these points demonstrate that our framework is both conceptually grounded and empirically validated for the LLM setting. We are happy to add a clarifying discussion in the final version and are confident that this work provides a strong, unified foundation for future comparative studies. We thank the reviewer for their careful consideration and hope these clarifications address any remaining reservations.
>
> We hope these clarifications meet your expectations and look forward to your positive reassessment.

---

> > ### Comment · Reviewer_aEWT · 2025-08-06
> > **Official Comment by Reviewer aEWT**
> >
> > I would like to thank the authors for a thorough rebuttal. After careful consideration of your detailed responses, I have raised my score. My initial concerns have been largely addressed.
> >
> > The core strength of this paper lies in its originality and significance. By introducing an interpretable, quantitative metric for strategic reasoning based on a well-grounded behavioral model, you have provided the community with a powerful new tool. This work effectively tackles the "what" and "why" of LLM strategic behavior, which is a necessary precursor to addressing the "how" of improving it. I now see the framework as a complete contribution on its own merits.
> >
> > While I still see avenues for future enhancement, particularly in a deeper analysis of the γ parameter and empirical model comparisons, I place greater weight on the foundational significance of the work itself.
> >
> > Thank you again for the productive and insightful discussion.

---

### Official Review · Reviewer_jSzb · 2025-06-30

**Clarity:** 3
**Significance:** 3
**Originality:** 4
**Rating:** 5
**Confidence:** 3

**Summary:**

This paper presents a framework to evaluate the strategic reasoning of LLMs. The authors use a concept from behavioral game theory called Truncated Quantal Response Equilibrium (TQRE) to assess how deeply LLMs think when playing different games. The study tests 22 LLMs to analyze their performance and reasoning styles. It also examines how assigning demographic personas to LLMs changes their strategic choices, highlighting potential biases in model behavior.

**Questions:**

**1**. You use the TQRE model to estimate reasoning depth. However, your conclusions about how a model reasons (e.g., using a "minimax" strategy) seem to come from analyzing the text generated by the model, not from the value itself. Could you explain the direct link between quantitative results with particular reasoning style?

**2**. In Section 4.2, you identify reasoning styles by analyzing excerpts from the models' outputs. This process can be subjective, as it depends on the researcher's interpretation of selected text. Did you use any methods to ensure the consistency and reliability of this analysis?

**3**. You observe that shorter reasoning chains can lead to better performance, which is an interesting finding. However, this analysis appears to be focused on only three top-performing models. How can you be sure this conclusion is generalizable to a wider range of LLMs? The current evidence seems insufficient to support such a broad claim.

**4**. The TQRE model estimates two parameters: reasoning depth $\tau$ and decision noise $\gamma$. The paper focuses entirely on the former. Could you please clarify why the $\gamma$ parameter was excluded from the analysis? Does this parameter provide important information about a model's consistency in decision-making?

**Ethical Concerns:**

["NO or VERY MINOR ethics concerns only"]

**Final Justification:**

This paper's core strength lies in its originality and significance, as it introduces an interpretable metric for strategic reasoning. Given that my key reservations have been thoroughly addressed, I now believe the reasons to accept this paper outweigh any remaining limitations.

**Limitations:**

yes

**Paper Formatting Concerns:**

There is no major formatting issue.

**Quality:**

3

**Strengths And Weaknesses:**

The main strength of this work is its novel approach to evaluating LLMs. Instead of using simple benchmarks, it applies a structured model from behavioral game theory. The method allows the authors to measure a specific metric, reasoning depth $\tau$, which is effective to quantify strategic thinking. The investigation into how demographic personas affect LLM decisions is also a significant and timely contribution.

However, the paper has several key weaknesses. First, the connection between the quantitative $\tau$ and the qualitative findings (the described reasoning styles) is not clear. The paper does not sufficiently explain how the TQRE model's outputs directly support the conclusions about how models reason. Second, the qualitative analysis of reasoning styles, such as "minimax," is based on selected model outputs, which introduces potential subjectivity. Finally, some of the broader claims, such as those about the effectiveness of Chain-of-Thought (CoT) prompting, are based on a small sample of top-performing models and may not apply to all LLMs.

---

> ### Author Rebuttal · Authors · 2025-07-31
>
> Thank you for the review. Below are our responses to your questions and comments.
>
>
> **Question 1 and Weakness 1 (about the link between quantitative reasoning depth and qualitative reasoning styles) Response:**
>
> This is a very insightful question. Our submission in this “social and economic aspects of machine learning” track has adopted a combined method that is typical in social science research:
>
>     Quantitative layer: A data-driven method, let the data tell the story from the observed behaviors, therefore effectively capturing the underlying mechanism and patterns of the agent. Here, we use TQRE as a rigorous, behavioral metric to quantify how deeply each LLM appears to reason in strategic settings.
>
>     Qualitative layer: To complement the quantitative analysis, we mirror the “verbal protocol analysis”, which is a well-established qualitative method in the behavioral sciences. Specifically, each CoT trace generated by the model is treated as an “interview transcript,” and we conduct thematic coding using a pre‑defined rubric, which is a qualitative analysis method for systematically labeling and organizing recurring ideas or patterns in qualitative data, such as text or transcripts (Braun & Clarke, 2006), to identify reasoning styles.
>
> This qualitative layer is essential because, just as with human agents, numbers alone rarely reveal the full narrative or structure of reasoning, and a contextual “story” is needed to interpret the meaning of quantitative signals. The linkage, therefore, is methodological rather than algebraic: the quantitative TQRE analysis tells us how far the agent seems to think; the qualitative “interview” reveals how it uses that depth.
>
> Employing both layers is standard practice when behavioural‑decision research aims to connect statistical regularities with interpretive understanding, and it is precisely this synthesis that our framework contributes to LLM evaluation. We will revise the text to make it clearer how this synthetic approach works.
>
>
> **Question 2 and Weakness 2 (about the method to avoid subjectivity and keep reliability) Response:**
>
> Yes, thank you for asking. We took the following steps to ensure the robustness and avoid subjectivity: We collected a diverse set of reasoning traces from different models and anonymized them before analysis. The researchers who coded each CoT trace for reasoning style did so without knowing which model produced it, analogous to a “blind review” process. This helps to minimize potential post-hoc bias. We then used a pre-defined structured coding rubric to classify reasoning styles.
>
> Then we did the cross-validation to mitigate subjectivity. Multiple researchers independently coded overlapping subsets of the data, and we compared their classifications to check for consistency.
>
> **Question 3 Response:**
>
> Our conclusion is not necessarily that “shorter chains are always better”, but that the common assumption “longer CoT → better strategic reasoning” does not always hold true. Therefore, the main takeaway is that the researchers should measure, but not assume the effect of CoT length.
>
>
> **Question 4 Response:**
>
> In our analysis, we focused primarily on τ because it is the key metric for strategic sophistication in behavioral game theory, as it directly reflects the depth of recursive reasoning and belief modeling. The γ parameter (decision noise), on the other hand, primarily serves as a control variable. It accounts for contextual confounders within the experimental setup and helps ensure the robustness of our primary measure τ. γ did not provide a consistent, model-wide insight comparable to τ, and due to page limit, we didn’t focus on discussing it and left more space to make the major objective clearer. Nevertheless, we acknowledge the reviewer’s suggestion and will include a detailed summary of γ values in the appendix of the revised paper. For reference, we have provided a table of γ estimates here.
>
> | Scenario | claude-3 | deepseek 2.5 | deepseek r1 | gemini 2.0 | gemma | gpt 4o | gpt o1 | gpt o3 mini | internlm 2.5 | LLAMA 405B | Llama 8B | Qwen 32 B |
> | :--- | ---: | ---: | ---: | ---: | ---: | ---: | ---: | ---: | ---: | ---: | ---: | ---: |
> | competitive | 5.833 | 7.669 | 11.383 | 1.719 | 8.834 | 2.053 | 0.441 | 1.131 | 5.384 | 6.154 | 0.000 | 0.370 |
> | cooperation | 1.687 | 3.347 | 0.348 | 2.009 | 0.348 | 1.667 | 3.349 | 3.347 | 1.667 | 1.667 | 0.336 | 1.696 |
> | Incomplete-Bayesian-p05 | 5.000 | 5.001 | 0.000 | 5.001 | 0.031 | 0.846 | 5.001 | 5.001 | 0.000 | 0.025 | 0.337 | 6.609 |
> | Incomplete-Bayesian-p09 | 5.276 | 5.276 | 0.014 | 5.276 | 0.317 | 0.950 | 5.276 | 5.276 | 0.593 | 0.441 | 5.861 | 0.842 |
> | Incomplete-Signaling | 0.355 | 0.120 | 0.156 | 0.300 | 0.129 | 0.236 | 0.395 | 0.120 | 0.061 | 0.000 | 0.081 | 0.882 |
> | mix motive | 0.000 | 1.711 | 0.000 | 0.348 | 0.000 | 0.000 | 3.717 | 1.720 | 0.000 | 0.000 | 0.000 | 1.813 |
> | Sequential | 38.675 | 52.656 | 40.615 | 0.000 | 52.362 | 34.198 | 0.676 | 35.734 | 42.306 | 37.102 | 48.384 | 48.972 |
> |   |   |   |   |   |   |   |   |   |   |   |   |   |
>
> **Weakness 3 Response:**
>
> In illustrating how our unified framework combines qualitative and quantitative methods, we focused our deeper analysis mainly on top-performing models for several reasons: (1) these models consistently provide explicit internal reasoning steps (Chain-of-Thought traces), whereas many smaller or less capable models do not provide reasoning steps as an API feature; and (2) as top performers, their behavior offers stronger and more discernible features, making trends more interpretable. For example, while many intermediate models perform moderately across games, the three models we selected demonstrate clear strengths and weaknesses (e.g., DeepSeek R1 excels in cooperative games but underperforms elsewhere), making it more intuitive to link reasoning patterns to their successes and failures. However, studies and results about lower-performing models are also included in the paper in section 4.3. We will elaborate more about those parts in the final version.
>
> We do not claim that the observed effects of CoT prompting are universal across all LLMs; indeed, each model exhibits its own reasoning style and behavioral pattern. Rather, our goal is to demonstrate that the unified framework can reveal these trends within individual models, particularly where reasoning traces are available and informative. We will clarify this point and its limitations in the revised manuscript as well.
>
> **Reference:**
>
> Braun, V., & Clarke, V. (2006). Using thematic analysis in psychology. Qualitative Research in Psychology, 3(2), 77–101.

---

> > ### Comment · Reviewer_jSzb · 2025-08-02
> >
> > Thank you for the detailed rebuttal. I now have a deeper understanding of your work. I agree that combining quantitative data with qualitative analysis is a standard approach in social science. I also accept your revised claim that longer reasoning chains are not always better.
> >
> > However, I still have two main concerns.
> >
> > **Role of the $\\gamma$ parameter**
> > - You treat it as a simple control variable, like a noise term. Yet, in the TQRE model, $\\gamma$ directly influences how much an agent responds to payoff differences. This suggests it has a behavioral meaning, such as a model's sensitivity to incentives.
> > - I appreciate the table of $\\gamma$ values provided in the rebuttal. However, the paper would be stronger if it also provided an intuitive explanation of what different magnitudes of $\\gamma$ imply about the model's behavior. For instance, does a
> > high $\\gamma$ consistently lead to more predictable or rational choices across different game types?
> >
> > **Suitability of the Level-k model for LLMs**
> > - (To be clear, this question is not intended to dismiss the manuscript's arguments. Rather, I am seeking a better understanding to help finalize my assessment.)
> > - The Level-k model assumes that strategic thinking happens in discrete, sequential steps. However, LLMs might operate differently. As powerful pattern recognizers, they could make a direct "jump" to a conclusion using heuristics learned from vast data, rather than iterating through levels of reasoning. A model based on iterative steps may not fully capture this aspect of LLM decision-making.
> > - How does the literature on bounded rationality in human agents address this potential discrepancy between iterative models and heuristic-based reasoning? Discussing this could strengthen the paper's foundation and address a potential concern for other reviewers.
> >
> > Thank you for the rebuttal, and I will wait for your response.

---

> > > ### Author Response · Authors · 2025-08-03
> > > **Further discussion with Reviewer jSzb**
> > >
> > > Dear Reviewer jSzb,
> > >
> > > Thank you for your response and further comments. We truly appreciate your engagement and we enjoy the discussion with you.
> > >
> > > 1. Regarding $\gamma$ and its role:
> > >
> > > If we observe an agent making a lot of "bad" or unexpected choices, it can be difficult to distinguish whether this is due to (1) shallow reasoning (low $\tau$) or (2) inconsistency/noise in execution (low $\gamma$), which itself could be influenced by many factors (risk preferences, individual behavioral quirks, or other unobserved context).
> > >
> > > I know in some literatures, people call gamma "payoff sensitivities", this is because in the mathematical function for quantal response, $\gamma$ directly multiplies the payoff. “payoff sensitivity” is technically correct, but in psychology and behavioral econ, this phrase can sound like “how much the agent cares about payoffs” (a preference), rather than a response process (what it should be).
> > >
> > > That’s why you (and many others! and myself when I first read these!) find “payoff sensitivity” a bit ambiguous. It’s not about their actual valuation of payoffs, but about how deterministic/noise their choices are given those payoffs.
> > >
> > > Importantly, $\gamma$ can be influenced by numerous contextual and individual factors and does not decompose cleanly into a single behavioral trait. It is therefore standard to include $\gamma$ as a necessary control to ensure $\tau$ estimates reflect reasoning depth, without being confounded by noise or context-driven unpredictability. This is why our conceptual and empirical focus remains on $\tau$, as the central and more interpretable signal of strategic sophistication in agentic behavior, particularly in LLM settings, where our main research question is about reasoning.
> > >
> > > We also agree that diving deep to $\gamma$, especially decompose the influential factors behind it, and detecting further relationships between $\gamma$ and other reasoning features would be very promising future directions. We would consider this in future work and will also add a discussion in our current paper.
> > >
> > > 2. About how LLM reasons and the suitability of reasoning model.
> > >
> > > This is a very fair concern, that’s also why we took a seriously look into LLMs' reasoning steps in section 4.2. We deeply analyzed the LLMs’ internal chain of thought outputs. We actually observed that they are thinking step by step by explicitly anticipating opponents’ possible actions, forming beliefs about those responses, and choosing accordingly. This qualitative evidence supports the use of TQRE and other level-k models as meaningful frameworks even for LLMs, not just humans. They merely recalling known solutions or “jumping” to answers, and found instead that their reasoning closely tracks the logic of recursive belief modeling, often verbalizing, step by step, how they arrive at a decision.
> > >
> > > Regarding to the literatures and research development paths in human agents, here is a brief literature review on empirical and theoretical works:
> > >
> > > (1) Empirical evidence of cognitive heterogeneity:
> > >
> > > Costa-Gomes, Crawford & Broseta (2001, Econometrica) show, using process-tracing, that some participants reason in steps, while others use simple rules or even guess.
> > >
> > > Johnson, Camerer, Sen & Rymon (2002, Cognitive Psychology) use “think-aloud” protocols and find people mix deliberative reasoning with fast, heuristic choices.
> > >
> > > Fischbacher & Gächter (2010, Econometrica) examine public goods games and find that both stepwise and heuristic behavior co-exist within and across individuals.
> > >
> > > Kreps (1998, European Economic Review) and Gigerenzer & Selten (2002, Bounded Rationality: The Adaptive Toolbox) provide further evidence and theory for the co-existence of fast heuristics and deeper reasoning.
> > >
> > > (2) theoretical models accommodate it:
> > >
> > > Nagel (1995, AER) and Camerer, Ho & Chong (2004, QJE) propose and validate level-k and cognitive hierarchy models as descriptive “as-if” frameworks, capturing the aggregate distribution of strategies, regardless of the cognitive path used.
> > >
> > > Stahl & Wilson (1994, Games and Economic Behavior; 1995, JEBO) empirically validate that agents mix steps and heuristics.
> > >
> > > Summary in short, in behavioral models we focus on here, heuristics are captured by “level-0” reasoning in level-k model families and specifically a very low $\tau$ in our model.
> > >
> > > (full reference list will be attached in the following comment due to space limitation)
> > >
> > > We hope these responses can address your concern, and in the meanwhile, we are also open to further discussions if you have any other comments or questions. Thanks!

---

> > > > ### Author Response · Authors · 2025-08-03
> > > > **reference list for the above official comment**
> > > >
> > > > Camerer, C. F., Ho, T.-H., & Chong, J.-K. (2004). A cognitive hierarchy model of games. Quarterly Journal of Economics, 119(3), 861–898.
> > > >
> > > > Costa-Gomes, M. A., Crawford, V. P., & Broseta, B. (2001). Cognition and behavior in normal-form games: An experimental study. Econometrica, 69(5), 1193–1235.
> > > >
> > > > Fischbacher, U., & Gächter, S. (2010). Social preferences, beliefs, and the dynamics of free riding in public goods experiments. American Economic Review, 100(1), 541–556.
> > > >
> > > > Gigerenzer, G., & Selten, R. (2002). Bounded rationality: The adaptive toolbox. MIT Press.
> > > >
> > > > Johnson, E. J., Camerer, C., Sen, S., & Rymon, T. (2002). Detecting failures of backward induction: Monitoring information search in sequential bargaining. Cognitive Psychology, 44(3), 178–218.
> > > >
> > > > Kreps, D. M. (1998). Anticipated utility and dynamic choice. In J. J. Laffont (Ed.), The New Palgrave Dictionary of Economics (pp. 66–89).
> > > >
> > > > Nagel, R. (1995). Unraveling in guessing games: An experimental study. American Economic Review, 85(5), 1313–1326.
> > > >
> > > > Stahl, D. O., & Wilson, P. W. (1994). Experimental evidence on players' models of other players. Journal of Economic Behavior & Organization, 25(3), 309–327.
> > > >
> > > > Stahl, D. O., & Wilson, P. W. (1995). On players' models of other players: Theory and experimental evidence. Games and Economic Behavior, 10(1), 218–254.

---

> > > > > ### Comment · Reviewer_jSzb · 2025-08-05
> > > > >
> > > > > I would like to thank the authors for a thorough and convincing rebuttal. My initial concerns have been addressed, at least more than a standard rebuttal, and I have raised my score accordingly.
> > > > >
> > > > > This paper's core strength lies in its originality and significance, as it introduces an interpretable metric for strategic reasoning. Having thoroughly addressed my key reservations, I now believe the reasons to accept this paper outweigh any remaining limitations. This paper provides the "what" and "why" of LLM strategic behavior, which must come before the "how" of fixing it. Therefore, I see the framework as a complete and high-impact contribution on its own.
> > > > >
> > > > > I acknowledge the valid concerns raised by other reviewers regarding the absence of novel algorithmic solutions that build upon the proposed framework. However, I place greater weight on the foundational significance of the work itself in my final evaluation.
> > > > >
> > > > > Although I am updating my evaluation, I am still open to hearing other reviewers' discussions, which can strengthen their points.

---

### Official Review · Reviewer_7W8V · 2025-07-01

**Clarity:** 3
**Significance:** 2
**Originality:** 3
**Rating:** 4
**Confidence:** 2

**Summary:**

This paper brings together two compelling threads—large language models (LLMs) and Behavioral Game Theory—by proposing a unified evaluation framework to assess strategic reasoning and social responsiveness. The authors validate their framework through a broad suite of experiments and conclude with a discussion of limitations and broader impacts.

**Questions:**

1. Given the identified alignment and reasoning-depth gaps, do you plan to introduce targeted innovations —such as alignment fine-tuning or auxiliary loss functions—to address these challenges in future work?
2. Considering that the contribution centers on proposing and validating an evaluation framework (without novel algorithmic contributions), have you considered submitting to NeurIPS Datasets & Benchmarks track instead of the main conference?

**Ethical Concerns:**

["NO or VERY MINOR ethics concerns only"]

**Final Justification:**

Considering the comprehensiveness of the empirical study and authors' justifications during rebuttal, I would like to increase my rating. But to be honest, I am not so confident at my evaluation so I will lower my confidence score.

**Limitations:**

yes

**Quality:**

2

**Strengths And Weaknesses:**

## Strengths ：
1.The proposed evaluation framework offers a unique bridge between LLM-based reasoning and behavioral game-theoretic analysis. Its theoretical foundations are well articulated, and the framework’s structure shows promise for applications in agentic strategic assessment.
2.The experimental design is comprehensive: multiple game settings, a diverse range of contemporary models, and fine-grained metrics are employed. This breadth ensures that the framework captures both performance and timeliness in evaluating reasoning depth.
3.The Social Effect with Human is thoughtfully executed, providing empirical insights.

## Weaknesses：
1.The paper reads like a combination of two largely independent studies, an evaluation framework and a human-centered social experiment. The narrative would benefit from tighter coupling, illustrating how the evaluation results directly inform or motivate the social effect findings.
2.While the experimental findings are well presented, the paper lacks proposed innovations to address observed shortcomings.
3. Since NeurIPS has a benchmark track, I am wondering if such paper should be submitted to the main track.

---

> ### Author Rebuttal · Authors · 2025-07-31
>
> Thank you for the review. Below are our responses to your questions and comments.
>
>
>
>
> **Weakness 1 Response:**
>
> The paper presents a two-stage study, but not just a combination of two separate experiments. The “social effect” study is not an independent effort, but a direct extension and application of our evaluation framework to real-world, human-centered challenges. It should be viewed as the second stage of the same framework:
>
>
>     Stage 1: We use TQRE to estimate each model’s intrinsic reasoning depth, establishing a behavioral baseline.
>     Stage 2: We enhance the estimation through introducing demographic personas or human factors, allowing us to isolate and diagnose how social cues shift model reasoning and expose demographic biases
> This staged approach demonstrates that an effective evaluation framework must move beyond quantitative performance alone to explicitly address the social and ethical dimensions of agentic reasoning.
> We can further clarify this linkage more accessible in a revision, making explicit how the social study is motivated by and only made possible through the unified framework introduced earlier.
>
>
> **Weakness 2 Response:**
>
>
> We adapt TQRE from behavioral economics and show that our framework completely subsumes NE and CH baselines. We emphasize that methodological innovation at this level of breadth and rigor constitutes a major scientific contribution. To our knowledge, no prior work has introduced a comprehensive methodology and extensive quantitative evaluation of LLM reasoning through the behavioral decision-theory lens. Our framework addresses major shortcomings of previous work, such as restrictive or unrealistic assumptions, by offering a rigorous, behavioral approach that directly quantifies reasoning depth and social effects.
> The issue of improving reasoning depth and alignment across diverse scenarios is a broad, ongoing challenge faced by the entire field, including leading research labs and companies. Our work provides, for the first time, a rigorous and quantitative behavioral lens to measure and compare reasoning depth, laying the necessary foundation for any principled model improvements in the future.
>
> By exposing and quantifying these limitations in a systematic and interpretable way, our framework enables future work on targeted interventions (e.g., model fine-tuning, training data augmentation, loss function design). We explicitly outline these next steps in our discussion section. Thus, we view our methodological innovation and actionable diagnostics as a crucial advance, enabling and accelerating broader progress on the open problems highlighted by the reviewer. Finding a single solution that aligns all models with one particular reasoning process or style remains a significant challenge. Aligning with one direction sometimes means misaligning with other directions; our framework’s value shows up at where the alignment or misalignment is, but aligning the model with a corresponding target is out of the scope of this work. But of course, we are looking forward to designing methodologies to achieve the ultimate alignment in the future based on the foundation established by this work.
>
>
> **Weakness 3 Response:**
>
> Our work goes beyond a dataset or static benchmark by providing new metrics and theoretical synthesis: it introduces a theoretically grounded, behavioral game-theoretic approach for analyzing LLMs as reasoning agents, including both the modeling methodology and actionable diagnostic tools for both developers and theorists. Moreover, we do not simply report scores, but interpret mechanisms of reasoning, model variation, and real-world alignment/fairness risks. This framework serves as a practical diagnostic tool that enables and facilitates researchers to iterate on LLM studies. These aligned with the NeurIPS main track’s goals of advancing both scientific understanding and practical impact in AI.
>
> Specifically, we submitted to the main NeurIPS track under the subsection **“Social and economic aspects of machine learning (e.g., fairness, interpretability, human-AI interaction, privacy, safety, strategic behavior)”**. Our work provides a methodological and empirical contribution at the intersection of strategic reasoning, human-AI interaction, and fairness in LLMs, all of which are central themes listed in this track. For these reasons, we believe our paper is a perfect fit for this subsection of the main track, rather than the datasets & benchmarks track.
>
>
> **Question 1 Response:**
>
> Yes, these are promising future directions. Our findings directly inform future directions in model improvement and lay the foundations for that.
>
> For example, the behavioral metrics produced by our framework allow for targeted fine-tuning to improve reasoning depth in specific game types, as well as the identification and mitigation of social biases via curated data augmentation. We are actively exploring these possibilities and believe our framework can serve as a foundation for iterative cycles of evaluation and model alignment. We see this as an important next step.
>
>
>
> **Question 2 Response:**
>
> This has been answered above in the weakness 3 response.

---

> > ### Comment · Reviewer_7W8V · 2025-08-09
> >
> > Thanks for the responses. Considering the comprehensiveness of the empirical study and your justification during rebuttal, I would like to increase my rating. But to be honest, I am not so confident at my evaluation so I will lower my confidence score.

---

### Note · Authors · 2025-08-12

Dear AC and Reviewers,

We sincerely thank the AC and reviewers for the constructive exchange. Our reviewers raised their scores after the rebuttal, recognizing the originality, significance, and interpretability of our framework, as well as its fit to the NeurIPS main track. We appreciate their acknowledgment that this work has solid theoretical and empirical evidence, therefore provides the “what” and “why” of LLM strategic behavior, offering a necessary foundation for further LLM post-training improvement.

During rebuttal, we clarified that the framework is a unified, staged design: Stage 1 measures intrinsic reasoning depth using and integration of quantitive and qualitative methods, Stage 2 extends this to social contexts, revealing demographic and situational shifts. This directly links methodological rigor with socially relevant findings. We provided complete tables of γ parameter, explaining its role as a control for robust τ estimation, and noting its potential for future behavioral analysis. We justified TQRE’s suitability through both theoretical grounding and qualitative Chain-of-Thought evidence showing step-by-step reasoning consistent with bounded-rationality models. The methodological link between quantitative and qualitative layers was made explicit, following established social science practice. We will add the promised clarifications and evidence in the final version.

Our large-scale evaluation of 22 LLMs across 13 games revealed stable, model-intrinsic reasoning styles, context-dependent shifts, and  effects of reasoning chain length. Reviewers emphasized the breadth, interpretability, and diagnostic value of these findings. We believe the concerns have been substantively addressed, and we are grateful for the feedback that has strengthened the paper and ensured its alignment with NeurIPS’s goals. We also thank the AC for their careful coordination throughout the process.

Thank you,

18134 Authors

---

### Decision · Program_Chairs · 2025-09-17

**Decision:**

Accept (poster)

**Comment:**

This paper proposes a behavioral game-theoretic framework to assess the strategic reasoning capabilities of large language models (LLMs). Drawing from the Truncated Quantal Response Equilibrium (TQRE) model, it estimates a reasoning depth parameter (τ) for 22 state-of-the-art LLMs across 13 one-shot strategic games, revealing variations in reasoning depth by model, game type, and context. Key findings include: models like GPT-o1-mini, GPT-o1, and DeepSeek-R1 exhibit higher reasoning depth; distinct reasoning styles (e.g., maximin, belief-based) emerge from chain-of-thought (CoT) analysis, where longer chains do not always correlate with better decisions; and demographic personas induce context-sensitive shifts, with some models showing improved reasoning under female identities and diminished performance under minority sexuality cues.

During review & rebuttal, reviewers raised concerns about: (1) the paper's fit for the main track versus benchmarks, addressed by authors emphasizing methodological innovation and social relevance; (2) unclear links between quantitative (τ) and qualitative (CoT styles) analyses, clarified via social science practices like thematic coding; (3) omission of γ analysis, responded to with a table and explanation of its control role; (4) TQRE's suitability for LLMs without model comparisons or fit metrics, justified theoretically as encompassing prior models, with fit data provided; (5) one-shot focus limiting adaptation insights, countered by noting decomposition of dynamics into one-shot decisions.

Both authors and reviewers engaged actively, with most reviewers raising scores post-discussion.

Overall, I think this paper introduces a rigorous, interpretable framework that advances understanding of agentic behavior, with strong empirical support and implications for fairness and alignment—core to the social aspects track. While not proposing fixes, it provides essential "what" and "why" insights for future improvements.